# Design of a methotrexate-controlled chemical dimerization system and its use in bio-electronic devices

Zhong Guo[1,2,3], Oleh Smutok[4], Wayne A. Johnston[2,3], Patricia Walden[2,3], Jacobus P. J. Ungerer[5,6], Thomas S. Peat[7], Janet Newman [7], Jake Parker[2,3], Tom Nebl [7], Caryn Hepburn[8], Artem Melman[4], Richard J. Suderman[9], Evgeny Katz[4] & Kirill Alexandrov [1,2,3,10,11✉]

Natural evolution produced polypeptides that selectively recognize chemical entities and their polymers, ranging from ions to proteins and nucleic acids. Such selective interactions serve as entry points to biological signaling and metabolic pathways. The ability to engineer artificial versions of such entry points is a key goal of synthetic biology, bioengineering and bioelectronics. We set out to map the optimal strategy for developing artificial small molecule:protein complexes that function as chemically induced dimerization (CID) systems. Using several starting points, we evolved CID systems controlled by a therapeutic drug methotrexate. Biophysical and structural analysis of methotrexate-controlled CID system reveals the critical role played by drug-induced conformational change in ligand-controlled protein complex assembly. We demonstrate utility of the developed CID by constructing electrochemical biosensors of methotrexate that enable quantification of methotrexate in human serum. Furthermore, using the methotrexate and functionally related biosensor of rapamycin we developed a multiplexed bioelectronic system that can perform repeated measurements of multiple analytes. The presented results open the door for construction of genetically encoded signaling systems for use in bioelectronics and diagnostics, as well as metabolic and signaling network engineering.

[1] ARC Centre of Excellence in Synthetic Biology, Sydney, NSW, Australia. [2] Centre for Agriculture and the Bioeconomy, Queensland University of Technology, Brisbane, QLD 4001, Australia. [3] School of Biology and Environmental Science, Queensland University of Technology, Brisbane, QLD 4001, Australia. [4] Department of Chemistry and Biomolecular Science, Clarkson University, 8 Clarkson Ave., Potsdam, NY 13699, USA. [5] Department of Chemical Pathology, Pathology Queensland, Brisbane, QLD 4001, Australia. [6] School of Biomedical Sciences, University of Queensland, Brisbane, QLD 4072, Australia. [7] Biology Group, Biomedical Manufacturing Program, CSIRO, 343 Royal Parade, Parkville, VIC 3052, Australia. [8] Waters Australia Pty Ltd, 38-46 South Str, Rydalmere, NSW 2116, Australia. [9] Nectagen, Inc., 2002 W 39th Ave, Kansas City, KS 66103, USA. [10] CSIRO-QUT Synthetic Biology Alliance, Brisbane, QLD 4001, Australia. [11] Centre for Genomics and Personalised Health, Queensland University of Technology, Brisbane, QLD 4001, Australia. ✉email: kirill.alexandrov@qut.edu.au

A protein's ability to selectively recognize small molecules and convert these molecular recognition events into biochemical activities is at the core of the elastic response of biological systems to environmental and physiological changes. Biological systems evolved a broad array of soluble and membrane receptors that gain or lose biochemical activity in response to small molecule binding. A vast majority of them represent allosteric systems where ligand binding controls conformation or dynamics of the proteins, thereby changing their interaction with the environment. The mechanisms by which this is achieved vary vastly, ranging from rigid body intramolecular rearrangements to global (re-) structuring events[1]. Major efforts have been directed into understanding the underlying molecular mechanisms and devising approaches for engineering of artificial allosteric protein receptors[2]. Such receptors have numerous applications as diagnostic protein biosensors, signaling surface receptors, and ligand-regulated transcription factors controlling cell metabolism[3]. Not surprisingly, most progress in understanding and engineering allostery has been made with systems that can be tested using high throughput assays (transcription factors, antibiotic resistance, and fluorescent proteins). Efforts to engineer allosteric systems broadly fall into two categories: (a) protein designs where new or re-designed allosteric sites are coupled to the active sites through long range intradomain interactions[2], and (b) utilization of ligand-binding domains that pass their conformational changes onto reporter domains thereby controlling their activity[4]. The latter approach is more straightforward but requires ligand-binding domains that undergo large conformational changes. Ligand-binding domains of solute-binding proteins have been repeatedly used in such designs, and resulted in a range of exceptionally useful imaging biosensors[5,6]. However, systematic analysis of the available protein:ligand structures revealed that the fraction of complexes that undergo large conformational transitions sufficient to operate reporter domains is quite small[7]. The situation is even more complex in the case of xenobiotics where natural binding proteins may not exist at all. Another attractive alternative is to use ligand dimerized systems ternary complexes among two protein domains and a small molecule. The prototypical examples of such complexes include rapamycin-inducible FK506-binding protein (FKBP)/FKBP-rapamycin binding domain (FRB) and gibberellin-inducible GAI/GID1 complexes, amongst others[8]. These have been used extensively to design both intra- and intermolecular switches and shown to control numerous reporter systems both in vitro and in vivo[9]. The main drawback of this approach is the paucity of such complexes and underdeveloped approaches for their de novo construction. Chemically induced dimerization (CID) and protein biosensor development efforts have both approached this challenge by either creating semisynthetic systems where the signal is generated through competitive displacement of modified ligand, or through computational and selection-based engineering of genuine high affinity ternary complexes[10–12]. There are two fundamentally different ways how small-molecule operated CID systems may function. In the classical case of macrocyclic compounds, small molecules form a complex with one of the binding domains inducing its stabilization and creating a new composite binding interface that is recognized by the second binder[13] (Fig. 1a). In an alternative scenario, binding of the ligand induces conformational changes in the "anchor" binder, and the second binder displays high affinity to this conformation without directly forming contacts with the ligand (Fig. 1b). Pioneering experiments of the Koide group demonstrated that in the first scenario, the ternary complex gains both affinity and specificity[14]. This is not expected to occur in the latter case where "anchor" domain:ligand affinity and specificity will determine the behavior of the ternary complex.

In the present study, we sought to understand the key design parameters for artificial CID systems and the resulting biosensors. We therefore perform multiple directed evolution experiments starting from the small molecule in the context of different "anchor" domains. We demonstrate that the nature of the binding domains strongly influences the effectiveness of the selection process. We use the developed CID system to construct electrochemical protein biosensors and resulting sensory electrodes, demonstrating that artificial CID systems are powerful modules for engineering information relays.

## Results

De novo construction of small molecule:protein ternary complexes has multiple design input parameters that influence performance of the resulting systems. As explained in the introduction, ab initio development requires sequential selection of two binding modules, at least one of which must bind to the small molecule of choice directly. Such binders are often referred to as "anchor" domains and their structure and properties are expected to have a major, yet poorly understood, influence on the selection of the second binder and the properties of the ternary complex. We decided to test the influence of the "anchor" domain properties on the second binder selection process and the functional model of the CID (Fig. 1a). In order to be able to test the practical utility of the created CIDs and resulting systems, we surveyed the PDB databank for complexes of therapeutic drugs that require monitoring due to their toxicity[15]. We picked methotrexate (MTX)—a synthetic analog of folate that is a frontline drug used to control neoplastic proliferation and inflammation[16]. In the former case high dose MTX administration often results in toxicity, and delays in renal clearance may lead to significant side effects and permanent organ damage[17]. Therefore, administration of high doses of MTX is accompanied by its monitoring by immunochemical and mass spectrometric methods, and this data informs decisions on drug dosing or introduction of the rescue therapy[18]. A rapid and accurate Point-of-Care MTX monitoring system is expected to have a significant clinical utility particularly in settings where access to high end diagnostic equipment is limited.

Our selection focused on complexes where the small molecule was at least partially solvent exposed, thereby creating potential new binding modalities on the surface of the anchor domain. We also favored small MTX-binding domains that were likely to be easily expressed in E. coli and have better chances of successful incorporation into larger fusion proteins. Based on these criteria, we selected three proteins shown in Fig. 1c–e that include dihydrofolate reductase:MTX complex, thymidylate synthase:MTX complex and VHH domain:MTX complex. A biotinylatable AVI-tag was added on the C-terminus of dihydrofolate reductase and VHH or N-terminus of thymidylate synthase and these open-reading frames were co-expressed in E. coli with BirA biotin ligase. The resulting biotinylated proteins were purified to homogeneity (Supplementary Fig. 1).

**Selection of MTX-dependent binders of chosen anchor domains.** To develop protein-binding domains capable of selective recognition of the anchor domain:MTX complex, we performed phage display selection using the nano-CLostridial Antibody Mimetic Proteins (nanoCLAMP) phage display library CNL-2, which contains $1 \times 10^{10}$ variants with 4, 7, and 5 randomized residues in three adjacent binding loops V, W, and Z, respectively[19]. We chose the nanoCLAMP domain due to its small size (15 kDa), ease of recombinant production, and finally the lack of cysteine residues enabling its use in both oxidizing and reducing environments. The ability to reversibly disrupt

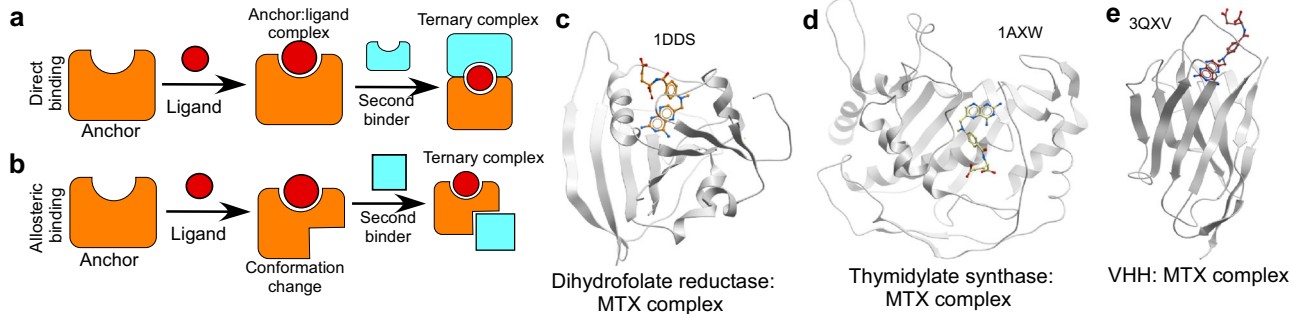

**Fig. 1 Development of small molecule dimerizing systems. a** Graphic representation of a dimerizing system that relies on formation of a new interface including an anchor protein and small molecule. The second binder recognizes this interface in a ligand-dependent way. **b** Dimerizing system where ligand binding leads to a conformation change of the anchor domain that is recognized by the second binder. **c** Ribbon representation of dihydrofolate reductase:methotrexate complex (PDB:1DDS). **d** thymidylate synthase:methotrexate complex (PDB:1AXW). **e** VHH: methotrexate complex (PDB:3QXV) displayed as white ribbon. Methotrexate is displayed in ball and stick representation with atomic colors.

nanoCLAMP:ligand interaction with polyol and chaotropic salt was also seen as a potentially beneficial feature[19]. The library was panned to enrich for binders selective for anchor domain:MTX complexes by preclearing the library for binders to the *apo*-form of the anchor domain prior to selection[19]. Following three rounds of panning against biotinylated dihydrofolate reductse, thymidylate synthase, and VHH in complex with MTX, 95 individual clones were randomly picked from the dihydrofolate reductase and VHH panning campaigns. The output from panning against thymidylate synthase did not produce enough clones for secondary screening and was not further analyzed. Binding of individual nanoCLAMPs to the biotinylated targets and the dependence of binding on the presence of MTX, was tested by soluble expression-based monoclonal enzyme-linked immunosorbent assay (semELISA), in which culture media from induced clones, which contain secreted nanoCLAMPs, are used as the primary binder (Supplementary Figs. 2A and 3A). To verify the results from the semELISA, we expressed in *E. coli* and purified the unique nanoCLAMPs and tested them in an inverse ELISA. The nanoCLAMP P2530 showed strong MTX-dependent binding at a DHFR concentration of 40 nM, while in the absence of MTX the signal was very close to background. However, at higher concentrations, this binder bound to DHFR in the absence of MTX (Supplementary Fig. 2B, C). This suggests that MTX contributes to complex formation, but the affinity differences may not be large. None of the other tested anti-DHFR clones produced clear MTX-dependent signals at the concentrations tested.

In contrast to thymidylate synthase and DHFR, 9 of the anti-VHH binders (Clones 1–9) demonstrated MTX-dependent binding (Fig. 2b, c and Supplementary Fig. 3) in both the crude semELISA and in the purified recombinant protein-based interaction assays. To confirm this further using an independent and more direct assay, we performed pulldown experiments capturing biotinylated VHH on streptavidin beads in the presence of purified recombinant nanoCLAMP clones. Interaction of all but clone 6 showed clear MTX dependence (Fig. 2d).

**Crystallization and structure solution of MTX:VHH:nano-CLAMP complexes**. The availability of several independent nanoCLAMP clones capable of selective MTX:VHH complex recognition prompted us to seek insights into the molecular mechanisms of ternary complex formation. To this end, we prepared purified ternary complexes of several VHH-nanoCLAMP constructs with different linkers designed to favor the formation of intramolecular or intermolecular complexes (Supplementary Fig. 4A). These complexes were subjected to crystallization trials and several of them produced crystals, often

with multiple crystal forms (Supplementary Fig. 4B, C). We were able to collect diffraction data sets for complexes containing nanoCLAMP3 and nanoCLAMP8 complex and solve the structure to 2.9 and 1.8 Å resolutionrespectively, using molecular replacement (Table 1). The overall structures of both complexes were very similar (Supplementary Fig. 5), and we therefore

**Table 1 Statistics of diffraction data and refinement.**

| Data collection | nanoCLAMP8-VHH | nanoCLAMP3-VHH |
|---|---|---|
| Wavelength (Å)[a] | 0.95372 | 0.95373 |
| Resolution (highest shell, Å) | 59.05−1.83 (1.88−1.83) | 48.38−2.9 (2.975-2.9) |
| Space group | C2221 | I2 |
| Cell constants (Å; °) | $a = 54.1$, $b = 95.9$, $c = 118.1$; $\alpha = \beta = \gamma = 90$ | $a = 173.2$, $b = 144.0$, $c = 181.8$; $\alpha = 90$, $\beta = 94.3$, $\gamma = 90$ |
| $V_M$ | 2.36 | 4.95 |
| Total measurements | 375,031(21,790) | 1,401,844(70,152) |
| Unique reflections | 27,424(1597) | 98,506(4830) |
| Average redundancy | 13.7 (13.6) | 14.2 (14.5) |
| $I/\sigma$ | 16.7 (3.5) | 7.5 (0.7) |
| Completeness (%) | 99.5 (96.5) | 100.0 (100.0) |
| Rpim | 0.036 (0.222) | 0.086 (1.210) |
| CC1/2 | 0.999 (0.952) | 0.994 (0.340) |
| *Refinement* | | |
| Resolution (highest shell, Å) | 1.83 (1.87-1.83) | 2.9 (2.975-2.9) |
| $R$[b] | 14.9(24.6) | 19.5(34.0) |
| $R_{free}$[c] | 20.2(42.6) | 24.2(38.7) |
| rmsd bonds (Å)/ angles (°) | 0.026/2.049 | 0.021/2.057 |
| *B-factor deviation* bonds/angles (Å²) | | |
| Main chain | 1.826/2.781 | 0.949/1.835 |
| Side chain | 3.874/5.405 | 2.421/4.204 |
| Residues in Ramachandran core (%)[d] | 98.17 | 91.70 |
| Protein atoms | 2200 | 14,529 |
| Solvent atoms | 830 | 3 |
| Ligand atoms | 35 | 245 |
| Average *B*-factor (Å²) | 19 | 73 |
| PDB accession code | 7RG7 | 7RGA |

[a]All data were collected at beamline MX1 or MX2 of the Australia Synchrotron (Melbourne, Australia)
[b]$R$ is the $R$-factor = $(\Sigma|F_O|-\Sigma|F_C|)/\Sigma|F_O|$.
[c]$R_{free}$ is the $R$-factor calculated using 5% of the data that were excluded from the refinement.
[d]Ramachandran core refers to the most favored regions in the $\varphi/\psi$-Ramachandran plot

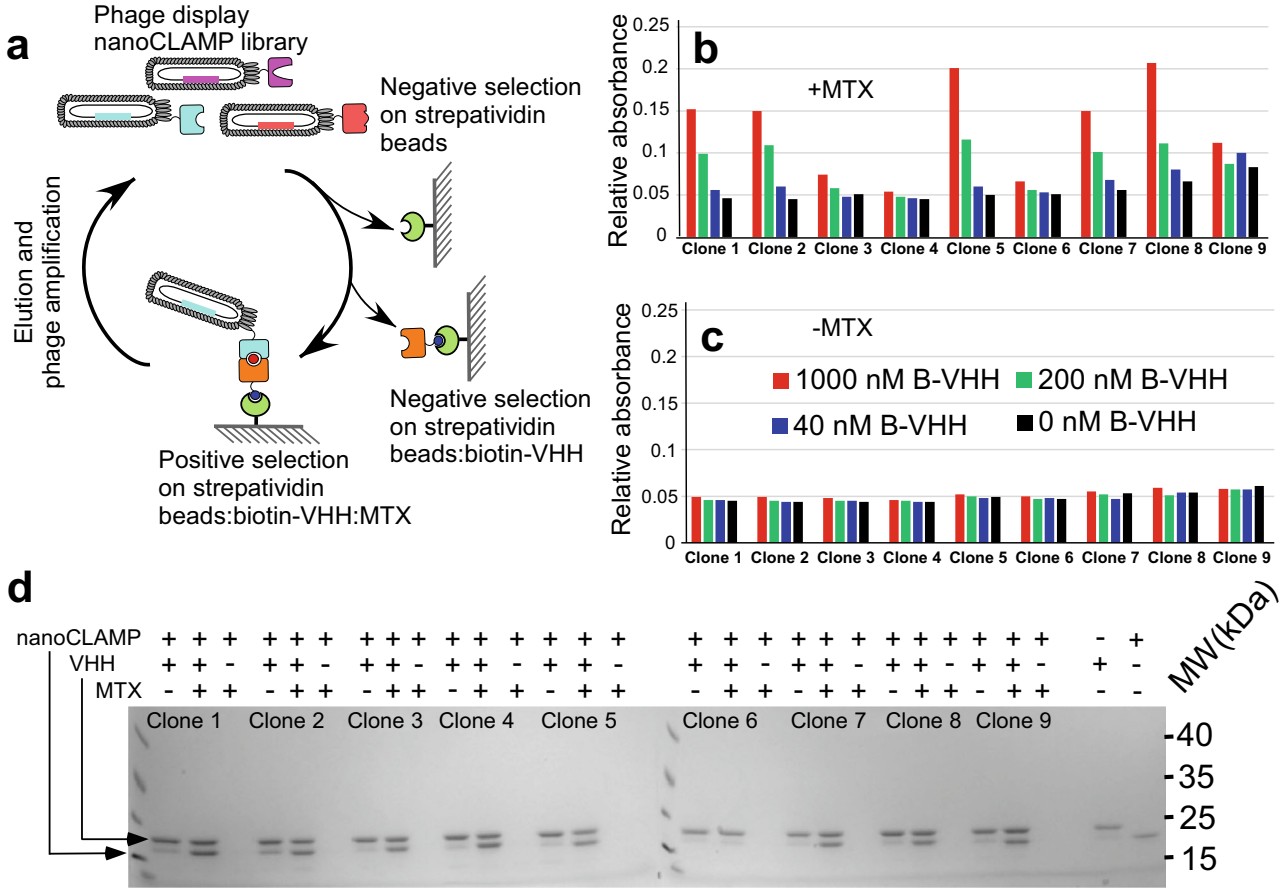

**Fig. 2 Selection of nanoCLAMP binders of VHH:MTX complex. a** Schematic representation of nanoCLAMP page library selection process aimed at identification of the binders selectively recognizing the VHH:MTX complex. **b** Inverse ELISA of recombinant purified nanoCLAMPs clones assessed for binding to VHH in the presence of indicated concentrations of MTX. Maleimide-coated plates were coated with purified nanoCLAMPs and incubated with three concentrations of biotinylated target protein and 10 µM MTX. The binding was detected using a Streptavidin–HRP-mediated colorimetric reaction. **c** As in (**b**) but without MTX. **d** SDS–PAGE analysis of the pulldown experiment where a mixture of biotinylated VHH domain and purified nanoCLAMP clones 1–9 were incubated with or without 10 µM of MTX and captured on the streptavidin beads prior to washing and elution with SDS–PAGE loading buffer. This experiment was performed once. Source Data are provided as a Source Data file.

performed a more detailed analysis on nanoCLAMP8 complex that was solved to a higher resolution. The overall structure of the complex is shown in Fig. 3a. The linker connecting the VHH and nanoCLAMP is disordered and is not visible in the structure, indicating that it is flexible and unlikely to contribute strain necessary to distort the structure of the complex. The well-defined density for MTX molecule is located in the cavity formed by CDRs 1, 3, and 4 of VHH domain (Fig. 3a, e). In the complex, the nanoCLAMP has a 700 Å$^2$ interface with the VHH:MTX complex that forms two interaction clusters. Cluster 1 involves interaction of invariable loop 64–71 with CDR1 and CDR4 regions of VHH (Fig. 3a–d). Cluster 2 is formed by the binding loops V, W, and Z that were randomized in the phage library with β-sheet 9 and 8 of VHH domain (nomenclature of structural elements is as in refs. [19,20]). This arrangement is quite surprising as one would expect the selection to result in structures where the variable elements of the nanoCLAMP would interact with the ligand or its surroundings. Comparison of VHH:MTX in complex with the nanoCLAMP with the structure of apo-VHH and VHH:MTX complex solved previously is shown in Fig. 3c. Binding of the nanoCLAMP introduced surprisingly little change in the structure of the VHH:MTX complex. From this it is clear that the rearrangement of the CDR1 region (Ser199-W206) of VHH upon MTX binding plays a key role in selective binding of the nanoCLAMP. The conformational change results in the 2.9 Å

movement of the C-alpha of Arg201 of CDR1 that forms two hydrogen bonds with Gly68 of Nanoclamp8 (Fig. 3f).

Comparison of the structure of the parental carbohydrate binding module (from which the library of nanoCLAMPs in this study was derived; PDB:2W1Q), with the nanoCLAMP structure in the solved complex, shows that the loop Lys64-Asp71 (colored in black in Fig. 3a) adopts a significantly different conformation in the ternary complex compared to the parental structure (Fig. 3d). This difference could be due to primary structure differences or be binding-induced. The wrapped loop in the apo nanoCLAMP structure corresponds to an extended loop in the complex with VHH:MTX structure that forms a range of hydrogen bonds with CDR1 and 4 of VHH (Fig. 3f). The adjacent loop 94–103 shows minor structural differences but it does not form new interactions in the VHH:MTX complex. In the apo-nanoCLAMP structure, two polar interactions are formed by Ser69 and Gly101, Lys72 and Pro99, respectively. In the structure with VHH:MTX, the above two polar interactions are absent and new hydrogen bonds are formed between Lys64 and Lys102, Asp71 and Gly101, Lys72 and Gly101, respectively. Variable loops V, W and Z also adopt different conformations but this is expected given the variation and their interactions with the VHH β-sheet. Given that the sequence of the apo-nanoCLAMP is not identical to that in the ternary complex, we cannot draw definitive mechanistic

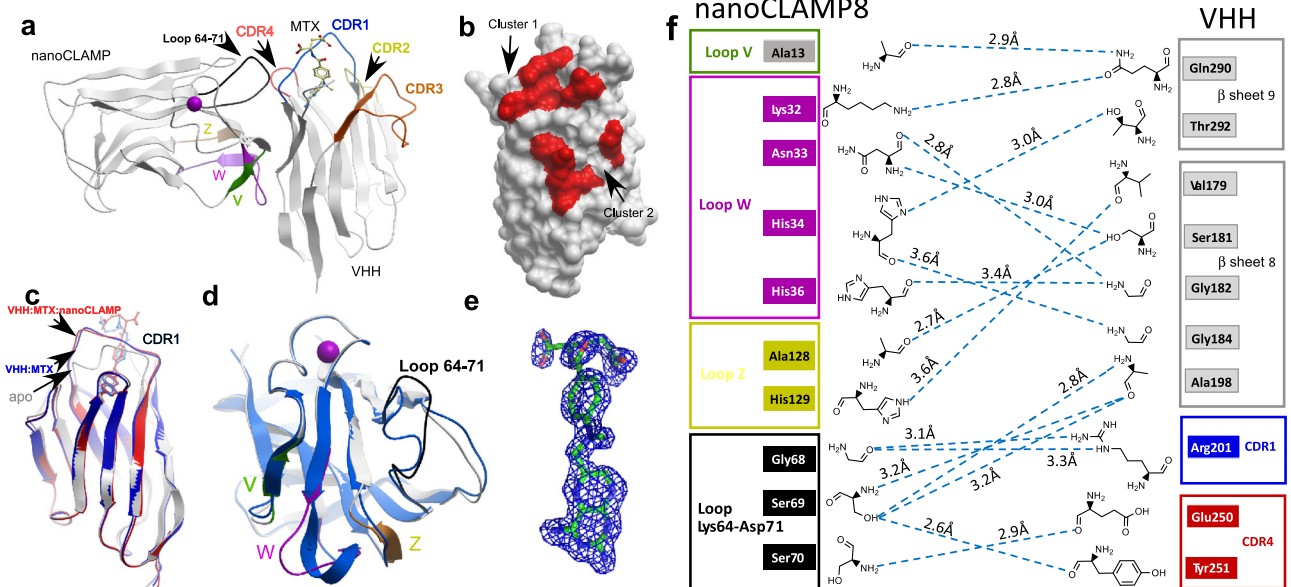

**Fig. 3 Structure of MTX:VHH:nanoCLAMP8 complex. a** Overview of the structure of the complex where protein molecules are displayed in gray ribbon and MTX in ball and stick representations. The magenta ball represents a calcium ion bound to the nanoCLAMP. The loops of the nanoCLAMP randomized in the library are colored in green (V), magenta (W) and yellow (Z). The invariable loop 64–71 is colored in black. **b** Surface representation of the MTX:VHH complex with red colored structural elements in <3.2 Å proximity to the nanoCLAMP. In the figure, VHH is turned 90° compared to (**a**). **c** superposition of the apo-VHH structure (PDB:3QXU) shown as gray ribbon with the structure of VHH:MTX complex (PDB:3QXV) displayed as blue ribbon and the structure of VHH:MTX from the ternary complex shown in red. Bound MTX molecules are displayed in the color of the respective complex. **d** Superposition of the *apo* form of the nanoCLAMP used a general library scaffold shown as gray ribbon with its structure from the nanoCLAMP from the ternary complex shown in blue with loops colored as in (**a**). **e** The omit map of MTX contoured at 3σ. **f** Graphic representation of the interface between the nanoCLAMP and VHH. The hydrogen bonds are shown as blue dashed lines and their lengths are displayed.

conclusions from this structural comparison. However, the overall topology of the nanoCLAMP is unlikely to undergo major change due to the substitutions present in the randomized V, W, and Z loops. As these are located at a distance from the Lys64-Asp71 loop, the observed conformational changes are likely to be related to the complex formation.

A remarkable feature of the interface is the abundance of polar interactions between the main chain on one molecule and side chains of another. In cluster 1, the nanoCLAMP backbone atoms bond with sidechains of CDR1 and 4 of VHH. The situation is reversed in cluster 2, where the nanoCLAMP donates predominantly side chains to the interaction while VHH contributes predominantly atoms of the main chain (Fig. 3e).

In the proposed interaction mechanism, the variable loops of the nanoCLAMP form a relatively weak binding interface with the constant region of VHH. Binding of MTX to VHH leads to restructuring of CDR1, pushing it out and rigidifying thereby providing an anchoring site for the invariable loop Lys64-Asp71. Therefore, MTX operates an allosteric cooperative switch where restructuring of the CDR1 loop plays the key role in increasing the ternary complex affinity.

Another remarkable feature of the solved structures is high conservation of the interaction mechanism. Although the variable loops of nanoCLAMP3 have very little similarity to that of nanoCLAMP8, the overall interface structure of its complex with VHH:MTX is remarkably similar (Supplementary Fig. 5A–C). While we cannot confirm that all of the selected clones operate using the same mechanism due to the lack of high-resolution structural information, the hydrogen–deuterium exchange experiments performed on clone 5 suggest that similar parts of the molecules are involved in complex formation and that interaction mechanism is conserved across the selected clones (Supplementary Fig. 6).

**Analysis of binding affinity between VHH and nanoCLAMP in the absence and presence of MTX.** In order to understand how the elucidated MTX-based interaction mechanism translates into equilibrium constants, we performed interaction analysis of the complex components using microscale thermophoresis[21]. For this we produced a recombinant version of the VHH N-terminally tagged with EGFP that could serve as an interaction reporter. When EGFP-VHH was titrated with increasing concentrations of MTX, we could observe a saturable signal increase that could be fitted to a $K_d$ value of $4.3 \pm 0.25$ nM which is identical to the affinity determined in the earlier study (Supplementary Fig. 7A)[22]. We then titrated EGFP-VHH with nanoCLAMP5 and fitted the data to the $K_d$ value of $3.7 \pm 0.23$ μM (Fig. 4a). When the experiment was repeated in the presence of the saturating concentrations of MTX, the affinity of the complex increased to $8.2 \pm 0.43$ nM (Fig. 4b) indicating that the MTX-induced conformational change in the VHH domain results in nearly three orders of magnitude increase in affinity between the protein components of the complex. This is consistent with the cooperative binding mode revealed by structural analysis and provides guidance for the design of efficient molecular switches.

**Construction and performance testing of a MTX biosensor.** The large change of the binding affinity between VHH and the selected nanoCLAMP domains upon binding of MTX enables utilization of this chemical-induced dimerization system for bioengineering applications. To demonstrate this, we took advantage of the artificial allosteric protein biosensor platform based on an engineered version of PQQ-glucose dehydrogenase (PQQ-GDH)[23–25]. PQQ-GDH is the principal component of electrochemical point-of-care glucose monitoring systems, and biosensors based on this enzyme are compatible with

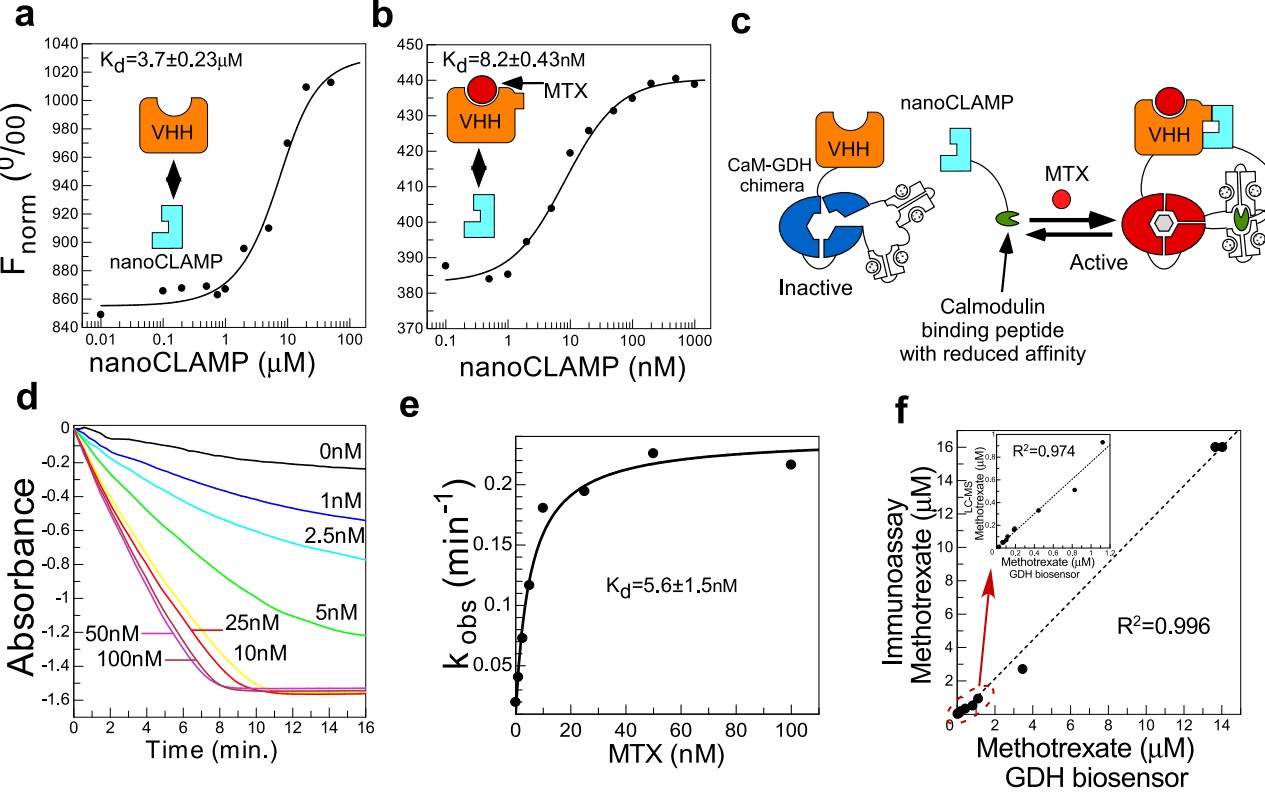

**Fig. 4 Biophysical analysis of MTX:VHH:nanoCLAMP complex interactions and its use to construct a methotrexate biosensor. a** Microscale thermophoresis analysis of the VHH interaction with nanoCLAMP5. In the experiment, 0.5 μM solution of EGFP-VHH was titrated with the increasing concentrations of nanoCLAMP5. Fit of the data led to a $K_d$ value of 3.7 μM. **b** As in (**a**) but using 10 nM solution of EGFP-VHH in the presence of 1 μM MTX. Fit of the data led to a $K_d$ value of 8.2 nM. **c** A schematic representation of CaM-GDH-based two-component MTX biosensor. Conformational change in the VHH domain enables binding of nanoCLAMP8-CaM-BP fusion that induces an activating conformational change in CaM-GDH. **d** Titration of a 10 nM solution of VHH-CaM-GDH and 100 nM of nanoCLAMP-CaM-BP with increasing concentrations of methotrexate. **e** Fit of the data from (**b**) to a quadratic equation leading to a $K_d$ value of 5.6 ± 1.5 nM. **f** Analyzing serum samples of patients receiving methotrexate therapy with the assay based on the developed MTX biosensor. The results of the assay were plotted against the values obtained using an Abbott Diagnostics immunochemistry station. The inset shows a separate plot for low concentration samples. Source Data are provided as a Source Data file.

commoditized disposable sensor electrodes[24]. Alternately, GDH activity can be easily monitored colorimetrically using redox dyes as reporters[26]. In our earlier work, we converted GDH enzyme into a switch module through its fusion with a calmodulin domain that keeps GDH in an inactive conformation. The chimeric molecules undergo a large increase in activity upon interaction with the calmodulin-binding peptide[23]. This switch module was used to create a two-component biosensor architecture where the ligand brings together the reporter and its activating peptide (Fig. 4c)[23,27]. To develop MTX biosensors, we constructed a fusion of anti-MTX VHH with CaM-GDH reporter. We also fused the selected nanoCLAMPs with strong MTX dependance (Fig. 2d) to a low affinity variant of calmodulin-binding peptide, and produced the resulting fusion polypeptides in recombinant form. The solutions of both recombinant proteins were mixed in the absence or presence of MTX, and the GDH activity was measured using the established colorimetric assay. All of the tested combinations displayed changes in the enzymatic activity in response to MTX (Supplementary Fig. 7B–H). For detailed characterization we selected the biosensor based on variant 8 that showed one of the best dynamic ranges under the tested conditions. As can be seen in Fig. 4d and e, titration of the biosensor component solutions with increasing concentrations of MTX led to a dose-dependent and saturable increase of GDH activity. The titration data enabled us to determine the $K_d$ value of 5.6 ± 1.5 nM for the interaction of MTX with the biosensor, which is close to the $K_d$ originally determined for MTX:VHH

interactions. The observed limit of detection was below 1 nM signifying an over 100 fold sensitivity improvement over the previously reported bioluminescent MTX biosensor[28].

To assess the potential practical utility of the developed biosensors we used them to quantify MTX in the serum of de-identified patients undergoing MTX treatment. Figure 4f shows that the assay based on our biosensor provides a quantitative readout with accuracy comparable to that of the immunochemistry method currently used by the reference diagnostic laboratory. The assay was able to accurately quantify MTX in samples across the nM–μM range of concentrations (Figs. 4f and inset, S8A, B).

**Construction of MTX sensing electrodes.** Solution-based electrochemical monitoring of glucose oxidation by GDH forms the technological base for endpoint and continuous glucose monitoring systems that have transformed the management of diabetes[29]. The demonstrated ability to create GDH-based biosensors to analytes such as ions, small molecules and proteins holds the promise of delivering a quantitative and low-cost biochemical analytic platform that leverages advancements in glucose monitoring technologies[23]. Its utility hinges on our ability to rapidly create biosensors to the desired analytes, and monitor their activities using electrochemical measurements. To further corroborate this, we tested performance of the developed MTX biosensor in an established solution chronoamperometric assay

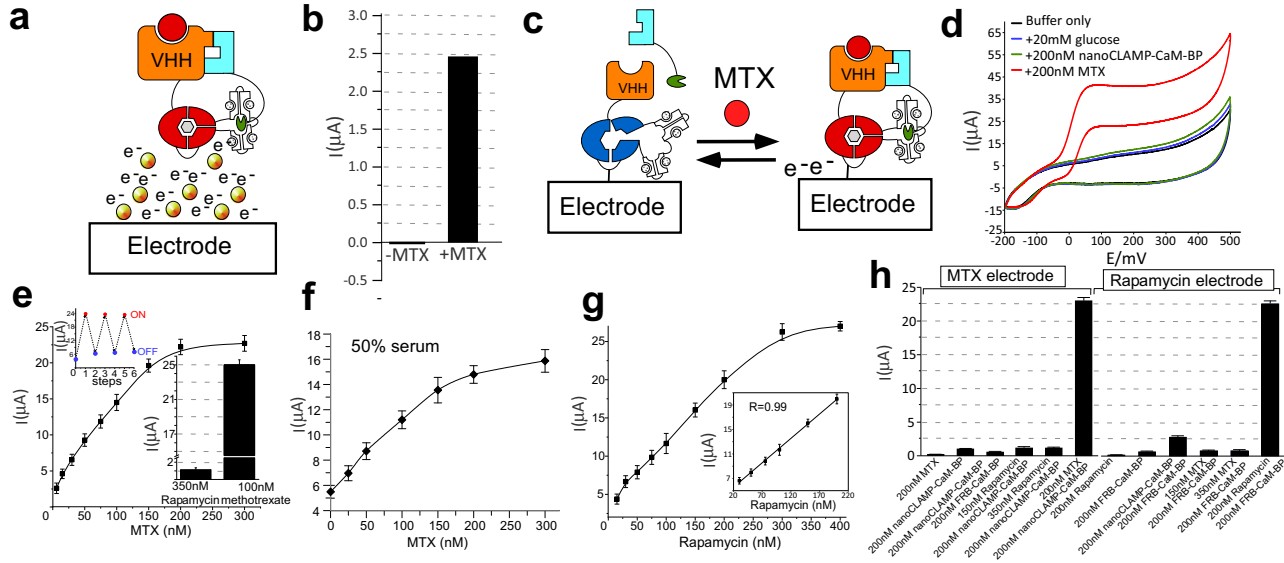

**Fig. 5 Electrochemical analysis of MTX biosensor and thereon based electrodes. a** A schematic representation of solution electrochemistry measurements where electrons generated by the MTX biosensor are transported to the electrode surface via an electron mediator (shown as golden balls). The functional elements are drawn as in Fig. 4c. **b** Electrochemical response of 0.5 µM VHH-GDH-CaM and 0.75 µM nanoCLAMP-CaM-BP solutions in the absence and presence of 1 µM of MTX. Sensor enzymatic activity is reported as maximum µA increase per minute using disposable DropSense gold electrode polarized at +0.1 V vs. Ag reference strip and mPMS as an electron transfer mediator. The data represents reading of individual electrodes tested sequentially. **c** A schematic of an electrode covalently modified with the VHH-GDH-CaM fusion component of MTX biosensor. The second component nanoCLAMP-CaM-BP is present in solution and associates with the electrode in the presence of the ligand leading to bioelectrode activation. **d** Typical cyclic voltammograms for MTX biosensor-based bioelectrode: black—buffer A solution (25 mM Tris–$H_2SO_4$ buffer, pH 7.2, 100 mM $Na_2SO_4$ and 1 mM $Ca(CH_3COO)_2$); blue—in presence of 20 mM glucose; green—in presence of 20 mM glucose and 200 nM nanoCLAMP-CaM-BP; red—in presence of 20 mM glucose, 200 nM nanoCLAMP-CaM-BP and 200 nM MTX. The electrode was scanned at the rate of 2 mV/s vs. Ag/AgCl/3 M KCl reference electrode at room temperature. **e** Increase in the electric current on the MTX bioelectrode shown in (**c**) following its exposure to the increased concentration of MTX. The upper inset shows reversibility of the activation after rinsing of the electrode. The lower inset compares the response of the electrode to the specific ligand MTX or unspecific ligand rapamycin. The data points represent average of three independent measurements performed on the same electrode. Data are presented as mean values ± standard error of mean. **f** as in (**e**) but using 50% human serum diluted in buffer A, 20 mM glucose and the indicated concentrations of MTX. **g** Rapamycin dose response of the bio-electrode constructed by cross-linking FKBP-CaM-GDH to the electrode and providing FRB-CaM-BP in solution. The inset provides a fit of the linear part of the titration curve. The data points represent average of three independent measurements performed on the same electrode. Data are presented as mean values ± standard error of mean. **h** Analysis of the MTX and rapamycin bio-electrodes for their ability to selectively recognize their cognate analytes in the presence of the component of the orthogonal GDH-biosensor. The bars represent values of average of three independent measurements performed on the same electrode. The error bars denote positive and negative boundaries of the standard error of mean. Source Data are provided as a Source Data file.

where 1-methoxy-5-methylphenazinium methyl sulfate (mPMS) acts as an electron transfer mediator between the enzyme in solution and the electrode (Fig. 5a)[23]. As expected, MTX induced a large increase in current produced by the MTX biosensor confirming its compatibility with solution electrochemical assays (Fig. 5b), and potentially with glucometer type disposable electrodes. However, the latter will require further optimization of protein stability as the manufacturing of such disposable electrodes often involves high temperature drying of the protein biosensor on the surface of the electrode (Supplementary Fig. 9 and the associated supplementary text).

A potential disadvantage of solution-based electrochemical assays is in the sensitivity limits set by the efficiency of electron transfer from the bio-element to the electrode[30]. In the case of glucose monitoring, the analyte is present at millimolar concentrations resulting in high electric currents. In many other diagnostic applications, including therapeutic monitoring of MTX, it may be necessary to quantify low nanomolar concentrations of the analytes which may be problematic for solution electrochemistry. One important advantage of PQQ-GDH is its ability to transfer electrons from the active site directly to electrodes provided that the latter is in close proximity[31]. As this can significantly improve the sensitivity of the system, we decided

to test if this approach can be applied to the developed MTX biosensor (Fig. 5c). Here we used a cysteamine/glutaraldehyde system to cross-link VHH-CaM-GDH in a random orientation to highly porous gold electrode through primary amines[32] (Supplementary Fig. 10B, C). We then exposed the electrode to the solutions of nanoCLAMP-CaM-BP and glucose in the absence or presence of MTX while monitoring the electrode using cyclic voltamperometry. As expected, addition of glucose and nanoCLAMP-CaM-BP resulted in a minor increase of current on the bioelectrode which is consistent with the background activation of the sensor. However, as can be seen in Fig. 5d, there was a large increase of current upon the addition of saturating concentrations of MTX, strongly suggesting that the solution-based component was able to interact efficiently and ligand-dependently with the immobilized GDH reporter component. We then titrated the electrode with MTX and observed a dose-dependent and saturable response to the drug (Fig. 5e). We further demonstrated that following activation, the electrode could be regenerated by a buffer rinsing procedure that removed the nanoCLAMP-CaM-BP and MTX, thereby resetting the activity of the electrode almost to the baseline level (Fig. 5e upper inset, SI). Given the concentrations of the components, the experiment represents an active site titration experiment that

could not be reliably fitted to a $K_d$, but demonstrates good linearity between 10 and 250 nM and a dynamic range of about 8-fold (Fig. 5b). The response was specific as no increase in current was observed when the electrode was exposed to high concentrations of rapamycin (Fig. 5e low inset). We finally tested the ability of the developed electrode system to respond to their cognate ligand in human samples. To this end, we repeated the experiments using MTX electrode and 50% human serum spiked with different concentrations of the drug. The electrode retained its functionality and responded to the presence of MTX in dose-dependent fashion, although with higher background signal (Fig. 5h and Supplementary Fig. 10E). Further work will be required to optimize the electrochemical parameters of the assay to reduce the background signal, improve the efficiency of electron transfer, as well as find ways of tackling the high viscosity of the sample (which complicates work with undiluted serum).

The presented results open an avenue for construction of electrochemical sensory electrodes to, at least in principle, any analyte. It also holds the potential to address one of the major challenges of distributed diagnostics and analytics—construction of multiplexed sensors. This is important as most biological states and health conditions cannot be unambiguously assessed with a single marker. To test the suitability of the developed platform to multiplexing we prepared a sensory electrode functionalized with a FKBP-CaM-GDH component of a rapamycin biosensor[23]. As in the case of the MTX biosensor, the activating component FRB-CaM-BP was supplied in solution. The rapamycin biosensor functionalized electrode performed very similarly to the above described MTX bio-electrode (Fig. 5f). We next tested the ability of the developed electrodes to operate in the presence of each other. As the GDH components of the biosensors are spatially segregated and the activating components present in solution are expected to be orthogonal to each other, multiple bio-electrodes should be able to operate in the same sample. As can be seen in Fig. 5g, both MTX and rapamycin bio-electrodes responded specifically to their cognate analyte despite the presence of the activating component for another biosensor. This confirms that they are functionally orthogonal despite shared architecture. This is an important step towards construction of electrochemical biosensor arrays based on cheap screen-printed gold electrodes.

## Discussion

In the current work we explored the design parameters for construction of small molecule-mediated protein dimerization systems and the resulting protein biosensors and bio-electrodes. We started our study on the assumption that it is possible to develop artificial protein complexes that are held together by an interface composed of the ligand and structural elements of both proteins. The nanoCLAMP binder selection campaign against three MTX:protein complexes yielded vastly different outcomes for each target, with one complex producing a large number of MTX-dependent binders, while the others produced little or none. While comparisons based on limited number of repeats must be treated with caution, the structural and biophysical analysis of the selected binders suggests that they exploit ligand-induced conformational changes in the anchor domain and do not form direct interactions with MTX. The crystal structure and HDX-MS analysis of MTX:VHH:nanoCLAMP ternary complexes revealed two interaction clusters on VHH:nanoCLAMP interface. The affinity measurements indicate that the binding of MTX increases the overall affinity of the complex by nearly three orders of magnitude through cooperative action of both interaction clusters. Remarkably, the segment of nanoCLAMP structure randomized in the library forms the interaction with the conformationally stable part of VHH, while the MTX-controlled

interaction is sensed by the invariable loop of the nanoCLAMP. It appears likely that the selection process drove positioning of the binder though the low affinity (probably mid-µM) interface in such a way that another weak interaction could form with the CDR1 loop rigidified through interaction with MTX. This mechanism helps to rationalize the differences in the outcomes in binder selection against different MTX:protein complexes. VHH displays the largest localized conformational change in its CDR1 loop upon ligand binding. It is followed by dihydrofolate reductase where MTX binding leads to a small change in the position of the α-helix over the binding site. In this case fewer ligand-dependent nanoCLAMPs were found and the difference in interaction strength with and without ligand was lower (Supplementary Fig. 3). The smallest conformation change upon MTX binding is detected in thymidylate synthase where no MTX-dependent binders were identified (Supplementary Fig. 11). Thymidylate synthase complex with MTX has the lowest affinity among complexes tested, which may have adversely affected the selection process[33]. However, we sampled only a relatively small number of clones and further screening may have resulted in identification of nanoCLAMPs with better selectivity for DHFR:MTX complex. Further, the success of the selection may also depend on the shape complementarity of the target and the binder, as well as the choice of the prevalent residues in the library that favor interface formation[34]. Combining the obtained and prior observations it appears that libraries of scaffold with randomized flexible loops are conducive to construction of small molecule binders and thereon based chemical dimerization systems. A number of studies previously developed antibodies to small molecules, as well as antibody recognizing antibody:small molecule complexes, indirectly supporting this assumption[12,35]. From the protein engineering standpoint, a single chain antibody such as VHH represents a more suitable scaffold than conventional antibodies. Recently Anticalins were shown to form high affinity complexes with a range of small molecules[36]. Given the lack of S–S bonds and substantial conformational changes of binding site forming loops, these proteins represent promising scaffolds for construction of chemically dimerized systems. Furthermore, in silico design coupled to diversity-based screening was recently used to construct a CID system regulated by farnesyl pyrophosphate, further expanding the number of avenues available for de novo construction of such systems[11]. The sophistication of the methods used still presents a challenge for their broad adoption, but constant methodological progress promises to address this limitation.

To demonstrate the practical utility of the developed artificial dimerization systems, we used them to develop GDH-based MTX biosensors and show that the resulting solution-based assays could detect low nM concentrations of the drug in human serum with accuracy comparable to that of clinically used diagnostic methods. The sensitivity and simplicity of the assay allows its transfer from complex immunochemistry stations onto more simple and ubiquitous clinical chemistry analyzers and plate readers. Such tests can be performed in smaller diagnostic laboratories and community clinics, thus increasing access to MTX monitoring and improving treatment outcomes.

We further took advantage of the GDH's ability to directly transfer electrons to conductive materials, in order to prototype sensory electrodes of MTX and rapamycin. We demonstrate that such electrodes can generate significant and dose-dependent currents in response to analyte exposure. It is expected that the efficiency of the electron transfer, and hence the sensitivity of the system, can be further significantly improved by attaching the biosensors to the electrode in an optimal orientation leading to improved electron transfer[37]. Importantly, we show that the electrodes are orthogonal to each other, and thus further

development of this approach is likely to enable quantification of multiple analytes in the same sample. This is an important feature that differentiates our approach from existing multiplexing efforts that are based on the use of unrelated redox enzymes that use different electrochemical readout schemes on each electrode[38]. Our approach potentially allows construction of sensory arrays furnished with biosensor variants displaying different affinities for the analyte. Such an array could cover a large concentration range that otherwise would require serial dilution of the sample. In the case of MTX monitoring this would significantly simplify testing, as during cancer treatment the circulating MTX concentration can fluctuate over several orders of magnitude (10 nM to >100 μM) and display large (>5 fold) interindividual variability[39]. Availability of high-resolution structure of nano-CLAMP:MTX:VHH complex simplifies construction of biosensor variants with reduced affinities for MTX. Furthermore, a similar approach can be used to construct biosensor arrays capable of rapidly assessing condition-specific marker panels.

One can also envisage that a similar approach can be utilized for construction of small molecule continuous monitoring systems that combine electrode immobilized GDH-CaM-Binder1 unit with soluble Binder2-CaM-BP. Such a system can be separated from the sample with a semi-permeable membrane, thus ensuring that the soluble component does not diffuse away and can associate with the electrode-bound reporter in a ligand-dependent manner. Use of such semi-permeable membranes in electrochemical biosensors is well established and would enable rapid transport of the analyte and circulating glucose to the biosensor[39]. Although fluctuations of glucose levels will influence the accuracy of this approach, it can be overcome by either reducing the $K_m$ of the GDH or by including a reference glucose electrode into the assembly.

In conclusion, our study identifies parameters that influence the development of chemically dimerizing systems and artificial switch systems based on them. While the current study focused on diagnostic and bioelectronic applications, the same approaches and design principles are applicable to construction of in vivo signaling and metabolic networks, intracellular communication systems and chemically controlled biomaterials.

## Methods

**Expression and purification of proteins**. Sequences of the MTX anchor protein domains were extracted from the following PDB files: dihydrofolate reductase:MTX complex (PDB:1DDS), thymidylate synthase:MTX complex (PDB:1AXW) and VHH:MTX complex (PDB:3QXV). The coding sequences optimized for *E. coli* expression were synthesized by Integrated DNA Technologies in frame with or without C-termina Avi-tag (Table S1) and cloned into pET28a(+) vector by the Gibson assembly method according to the manufacturer's instructions (New England Biolabs). The in vivo biotinylation of the anchor domains was performed by co-expression of AVI-tagged proteins with BirA biotin ligase (BirA chloramphenicol resistance plasmid: pBirACm (Addgene plasmid # 48307)). The anchor domains with or without AVI tag were overexpressed in *E. coli* BL21(DE3) cells. Cells were grown at 37 °C and expression was induced overnight at 18 °C in the presence of 0.3 mM isopropyl thio-β-D-galactoside. To produce the biotinylation form of proteins, 25 mg biotin were added to 1 L of bacterial culture during expression. Proteins were purified by Ni-NTA chromatography using standard protocol and followed by size-exclusion chromatography at 4 °C on Superdex G75 16/60 column (GE Healthcare) in the buffer containing 20 mM Tris–HCl, pH 7.2, 100 mM NaCl. The purified proteins were concentrated as required using Amicon Ultra 10K MWCO centrifugal filters (EMD Millipore, USA) and aliquoted fractions were stored at −80 °C after snap freezing in liquid nitrogen. The protein concentration of the purified proteins was determined using Nanodrop (Thermo Fisher Scientific). The concentration calculation was based on molar extinction coefficients at 280 nm calculated using Protparam online tool ExPASY (SWISS Institute of Bioinfomatics) using on the amino acid composition as an input. The extent of biotinylation was estimated pulldown analysis on streptavidin beads and was determined to be in the range of 75–80%. The SDS–PAGE analysis of the resulting proteins is shown in Supplementary Fig. 1.

**CNL-2 nanoCLAMP library panning**. Construction of CNL-2 phage library was described previously[19]. For the first round of panning, 3 L of 2× yeast extract-tryptone medium with 2% glucose and 100 mg/ml carbenicillin (2 × YT/2% glucose/100 μg/ml CAR) was inoculated with 4 ml of the CNL-2 library glycerol stock (OD600 = 75), to an $OD_{600}$ of approximately 0.1 and grown at 37 °C with 250 rpm agitation until the $OD_{600}$ reached 0.45. The library was infected by adding helper phage VCSM13 (Stratagene, Cat#200251) to 750 ml of culture at an MOI of 20 phage/cell, and incubating at 37 °C, 100 rpm for 30 min, then 250 rpm for an additional 30 min. The cells were pelleted at 10K × *g* for 10 min, and the media discarded. The cells were resuspended in 1.5 l 2 × YT medium with 100 μg/ml CAR, 70 μg/ml kanamycin (KAN), and incubated overnight at 30 °C, 250 rpm. The cells were combined, and 150 ml was centrifuged at 10K × *g* for 10 min. The phage containing supernatant was transferred to clean tubes and precipitated by adding 37.5 ml of 5× PEG/NaCl (20% polyethylene glycol 6000/2.5 M NaCl), and incubated on ice for 25 min. The phage was pelleted at 13K × *g*, 25 min and the supernatant discarded. The phage was resuspended in 10 ml 20 mM NaH₂PO₄, 150 mM NaCl, pH 7.4 (PBS), then centrifuged at 15K × *g* for 15 min to remove insoluble material. The phage was precipitated a second time by adding 1/4 volume 5× PEG/NaCl, incubated on ice for 5 min, and pelleted at 13K × *g*, 10 min at 4 °C. The phage pellet was resuspended in 3 ml PBS and quantified by absorbance at 268 nm ($A_{268}$ = 1 for a solution of 5 × 10¹² phage/ml)[13].

For each target protein, three sets of 100 μl of Dynabeads MyOne Streptavidin T1 (ThermoFisher Scientific) magnetic beads slurry were washed 2 × 1 ml with PBS-T (PBS with 0.05% Tween 20), applying magnet in between washes to remove the supernatant, and then blocked in 1 ml of 2% dry milk solution in PBS with 0.05% Tween 20 (2% M-PBS-T) for 1 h, rotating, at room temperature. The first set was used for preclearing the phage for binders to Streptavidin and the beads, the second set was used for preclearing the phage of binders to biotinylated target alone, and the third set was used for pulling down phage-target protein/methotrexate (MTX) complexes. To preclear the phage against beads alone, 1 ml of phage was prepared at a concentration of 2 × 10¹³ phage/ml in 2% M-PBS-T, and added to set-1 and incubated 1 h, rotating. While the phage was incubating with the first set of beads, the second set of beads (set-2) was saturated with biotinylated target protein at 400 nM in 1 ml 2% M-PBS-T for 30 min at room temperature, rotating (all incubations on beads henceforth at room temperature, rotating). The protein coated set-2 beads were applied to magnet and rinsed three times with 1 ml PBS-T, resuspending beads between each wash (all washes henceforth involved applying magnet and resuspending beads upon addition of each wash), to remove unbound protein. The precleared phage from set-1 beads was removed and applied to the protein-coated beads and incubated 1 h to remove phage that bound the protein alone. During this incubation, the third set of beads was loaded with biotinylated protein as described for set-2, except MTX was included at 10 μM in order for protein/MTX complexes to form. MTX was included in all incubations and washes from this point forward, until the phage was eluted. The phage was removed from the set-2 beads and transferred to set-3 beads, containing biotinylated protein/MTX complexes, to capture phage specific for protein/MTX complex, and incubated 1 h. Following capture on the beads, the beads were washed eight times with 1 ml PBS-T/MTX. The phage was then eluted with 0.8 ml 0.1 M glycine, pH 2 for 10 min rotating, the magnet applied, and the eluted phage aspirated into a clean tube containing 72 ml 2 M Tris base to neutralize, and then added to 9 ml of mid-log Xl1-Blue *E. coli* cells and incubated for 45 min, 37 °C and agitated at 150 rpm. The infected cells were expanded to 100 ml 2 × YT/2% glucose/ 100 μg/ml CAR and incubated overnight at 30 °C at 250 rpm.

The overnight cultures were harvested by measuring the $OD_{600}$, centrifuging the cells at 10K × *g* for 10 min and then resuspending the cells to an $OD_{600}$ of 75 in 2 × YT/18% glycerol. To prepare phage for the next round of panning, 5 ml of 2 × YT/2% glucose/100 μg/ml CAR was inoculated with 5 μl of the 75 $OD_{600}$ glycerol stock and incubated at 37 °C, 250 rpm until the $OD_{600}$ reached 0.5. The cells were superinfected at 20:1 phage:cell, mixed well, and incubated at 37 °C, 30 min, 150 rpm and then 30 min at 250 rpm. The cells were pelleted at 5500 × *g*, 10 min, the glucose containing media discarded and the cells resuspended in 10 ml 2 × YT/100 μg/ml CAR/70 μg/ml KAN and incubated overnight at 30 °C, 250 rpm.

The overnight phage prep was processed as described above. The phage was then prepared at $A_{268}$ = 0.8 in 2% M-PBS-T, and the panning and pre-clearing continued as described, except in the second and third rounds, the biotinylated target was coated on the set-3 beads at 100 and 10 nM, respectively, with 10 μM MTX. Washes after phage-capture on set-3 beads was also increased in the second and third rounds to 10 and 12 washes, respectively. In round 3, neutravidin-coated magnetic beads (Spherotech) were used in place of streptavidin-beads to reduce enrichment for streptavidin binders.

**Qualitative semELISA of individual clones following panning**. At the end of the last panning round, individual colonies were plated on 2 × YT/2% glucose/100 μg/ml CAR agar plates following the 45 min 150 rpm recovery at 37 °C of the infected XL1-blue cells with the eluted phage. The next day, 95 colonies were inoculated into 400 μl 2 × YT/2% glucose/100 μg/ml CAR in a 96-deep-well culture plate, and grown overnight at 37 °C, 300 rpm to generate a master plate, to which glycerol was added to 18% for storage at −80 °C. To prepare an induction plate for the ELISA, 5 μl of each master-plate culture was inoculated into 400 μl fresh 2 × YT/0.1% glucose/100 μg/ml CAR medium and incubated for 2.75 h at 37 °C, 300 rpm. IPTG was then added to 0.5 mM and the plates incubated at 30 °C with 300 rpm shaking overnight. Because the phagemid contains an amber stop codon, some nanoCLAMP protein is produced

without the pIII domain, even though XL1-blue is a suppressor strain, resulting in the periplasmic localization of some nanoCLAMP, of which some percentage is ultimately secreted to the media. The media can then be used directly in an ELISA assay (soluble expression-based monoclonal enzyme-linked immunosorbent assay: semELISA). After the overnight induction, the plates were centrifuged at $1200 \times g$ for 10 min to pellet the cells. Streptavidin-coated microtiter plates (ThermoFisher) were rinsed three times with 200 μl PBS, and then coated with biotinylated target proteins at 3 μg/ml with and without methotrexate (MTX) at 10 μM with 100 μl/well and incubated 1 h. For blank controls, a plate was incubated with 100 μl/well PBS, 10 μM MTX. Throughout the ELISA assay, all solutions on plates with MTX contained MTX at 10 μM to ensure that the complexes did not dissociate. The wells were then washed three times with 200 μl PBS-T (±MTX), and blocked with 200 μl 2% M- PBS-T (±MTX) for 1–3 h. The block was removed and 50 μl of 4% M-PBS-T (±20 μM MTX) added to each well. At this point 50 μl of each induction plate supernatant was transferred to the blank and protein-coated wells and pipetted 10 times to mix, and incubated 1 h. The plates were washed four times with 200 μl PBS-T (±MTX) and the plates dumped and slapped on paper towels in between washes. After the washes, 75 μl of 1/2000 dilution anti-FLAG-HRP (Sigma A8592) in 4% M-PBS-T (±MTX) was added to each well and incubated 1 h. The anti-FLAG-HRP was discarded and the plates washed as before. The plates were developed by adding 75 μl TMB ultra substrate (ThermoFisher), and analyzed for positive signals compared to controls. Positive clones were then grown up from the master plate by inoculating 1 ml 2×YT/2% glucose/100 μg/ml CAR with 3 μl glycerol stock and incubated for at least 6 h at 37 °C, 250 rpm. The cells were then pelleted and the media discarded. Plasmid DNA was prepared from the pellets using the Qiaprep Spin Miniprep Kit, and the sequences determined by Sanger sequencing at Genewiz (South Plainfield, NJ).

**Expression and purification of nanoCLAMPs for ELISA binding assay.** The nanoCLAMP cDNA of positive clones identified in the semELISA was amplified and cloned into a pET15b vector as described previously[1] using the In Fusion system (Takara). The vector includes an N-terminal 6-His tag and C-terminal 13 amino acid GS-linker followed by a cysteine. This construct codes for a mature protein of 163 amino acids with a calculated molecular weight of 17.6 kDa. Chemically competent NEc1 E. coli (Nectagen) were heat- shocked with 1.5 μl of the In Fusion reaction and recovered in 500 μl SOC for 1 h, 37 °C at 250 rpm. The cells were plated on 2×YT/2% glucose/100 μg/ml CAR agar and incubated overnight at 37 °C. Individual colonies were grown up in 10 ml of 2×YT/2% glucose/100 μg/ml CAR cultures for at least 7 h, after which plasmids were purified for sequencing to confirm the presence of the nanoCLAMP cDNA.

For 100 ml expression cultures, 100 ml 2×YT/100 μg/ml CAR cultures were inoculated to an $OD_{600}$ of 0.1 and grown to $OD_{600} = 0.7$–1.0 at 37 °C, 250 rpm. The cultures were cooled to around 30 °C, induced by adding IPTG to 0.5 mM, and incubated at 30 °C, 250 rpm for 24 h. Cells were pelleted at $10K \times g$, 10 min, the supernatant discarded, and frozen at −80 °C. The cells were thawed and lysed by adding 10 ml Qiagen Buffer A pH 8.0 (6 M GuHCl, 100 mM NaH$_2$PO$_4$, 10 mM Tris–HCl, adjusted to pH 8.0) + 1 mM TCEP by mixing well. The insoluble material was pelleted at $16K \times g$ for 20 min and the cleared supernatant added to 0.25 ml NiNTA-SF resin (Qiagen) equilibrated in same buffer, and incubated rotating for 2 h at room temperature. The beads were pelleted at $1K \times g$, 2 min, 15 °C, and the flow-through discarded. The resins were transferred to 2 ml disposable columns using Qiagen buffer A, pH 8.0, and washed as follows: 3 × 1 ml Qiagen buffer A, pH 8.0 + 1 mM TCEP, then 3 × 1 ml Qiagen Buffer A, pH 8.0 (no TCEP). The proteins were refolded on the resin by washing the resin 3 × 1 ml, then 2 × 5 ml, then 3 × 1 ml with 50 mM NaH$_2$PO$_4$, 300 mM NaCl, pH 8.0. The proteins were eluted with 5 × 0.2 ml 50 mM NaH$_2$PO$_4$, 300 mM NaCl, 250 mM imidazole, pH 8.0. The proteins were quantified by absorption at $A_{280}$ and buffer exchanged into 20 mM MOPS, 150 mM NaCl, pH 6.5 (MBS), and normalized to 1 mg/ml.

**Inverse ELISA of purified nanoCLAMPs to verify MTX dependence of binding.** Each purified nanoCLAMP, at 1 mg/ml in MBS buffer, was reduced with 1 mM TCEP in 50 ml for 30 min at room temperature. During the reduction, maleimide-coated microtiter strips (Thermo) were rinsed 3 × 200 μl PBS-T. The reduced proteins were diluted to 5 mg/ml in maleimide coupling buffer (MCB: 100 mM NaH$_2$PO$_4$, 150 mM NaCl, 10 mM EDTA, pH 7.2) in 1 ml, and then used to coat the plates at 100 μl/well overnight, 4 °C. For negative control well, plates were coated with MCB alone. The plates were rinsed 3 × 200 ml with PBS-T, inactivated with 100 mg/ml L-Cys in MCB for 30 min at 200 ml/well, rinsed 3 × 200 ml with PBS-T, and then blocked with 200 ml/well 2% M-PBS-T for 2 h at room temperature. The blocking solution was removed, the plates patted dry, and stored inverted at 4 °C with desiccant. The biotinylated target proteins were diluted in 2% M-PBS-T (±10 μM MTX) to 1000, 200, 40, and 0 nM and added to the nanoCLAMP-coated plates, and incubated for 1 h at room tempreture. The plates were washed 4× PBS-T (±10 μM MTX), then incubated with 1/2000 SA-HRP Ultra (Thermo ENN504), (±MTX) at 75 μl/well, for 1 h at room temperature. The plates were washed 4× again as before and then developed with TMB Ultra (Thermo).

**Pulldown analysis of MTX dependent interaction of VHH and nanoCLAMP clones.** For pulldown experiments 10 μl of Strep Tactin XT super flow beads (IBA

Solution for Life Science) in 1.5 ml Eppendorf tube were equilibrated and washed with buffer containing 20 mM Tris, pH 7.4, 300 mM NaCl, 0.2% Triton X-100. The supernatant was removed by centrifugation at 2000 rpm and 20 μl of 5 μM of VHH-AVI biotin protein was added to the drained beads. The protein was incubated in the presence or in the absence of five times molar excess of nanoCLAMPs with or without 10 μM MTX for 30 min. Then the beads were washed three times with 250 μl buffer containing 20 mM Tris, pH 7.4, 300 mM NaCl, 0.2% Triton X-100. Finally, the beads were eluted with 20 μl pre-heated SDS–PAGE protein loading buffer containing 200 mM DTT. After incubation at 95 °C for 15 minutes the samples were loaded onto SDS–PAGE for analysis.

**Spectrophotometric analysis of PQQ-GDH enzymatic activity.** The GDH-CaM contained enzymes were reconstituted with PQQ in 1:1.5 ratio. The GDH enzyme assay was performed as described previously[40]. Briefly, the 1.0 ml volume reactions comprised 20 mM glucose, 0.6 mM phenazine methosulfate (PMS), 0.06 mM 2,6 dichlorophenylindophenol (DCPIP), 20 mM Tris–HCl, pH 7.2, 100 mM NaCl, 1 mM CaCl$_2$ and enzyme were carried out in polystyrol cuvettes (SARSTEDT). The enzymatic assays were performed at 25 °C by monitoring the decrease in absorbance of DCPIP at 600 nM using a Cary 60 UV–VIS absorbance spectrometer operated by Cary WinUV Software.

The linear phase of the recorded curves was fitted as linear function to obtain $K_{obs}$. In order to obtain the $K_d$ of the interaction of GDH biosensor with its ligand, the $K_{obs}$ data were plotted against the concentration of the ligand and the data were fitted to the explicit solution of the quadratic equation (1) describing the $E + S <> ES$ binding equilibrium, where $K_d$ is defined as $K_d = [E] * [S] / [EL]$. $[E_0]$ and $[L_0]$ refer to the total enzyme and ligand concentration (free and bound) in the cuvette. Under these conditions the fluorescence is described by

$$F = K_{obs}(\min) + (K_{obs(\max)x} - K_{obs(\min)})/ * ([E_0] + [L_0] + K_d)/2$$
$$- ([E_0] + [L_0] + K_d)^2/4 - [E_0] * [L_0])^{1/2}/[L_0]$$

$K_{obs}$ represents the measured rate, while $K_{obs}$ (min) and $K_{obs}$ (max) refer to the minimal and maximal rates observed, respectively. A least-squares fit of the data to Eq. (1) using the software package Grafit 5.04 (Erithacus software) was used to extract the $K_d$ value.

**Crystallization and structure solution of MTX:VHH:nanoCLAMP complex.** The crystals of nanoCLAMP8:VHH:MTX complex were obtained by mixing 0.2 μl of protein at 18.5 mg/ml in 20 mM Tris–HCl, pH 7.4, 40 mM NaCl, 5 mM DTT with 0.2 μl of reservoir solution containing 15% (w/v) PEG 4000, 0.2 M DL-malate-imidazole pH 6.0, 0.1 M MgCl$_2$. The drop was set up in a sitting drop plate (SD2, SwissSci, UK) and incubated at 20 °C against 50 μl reservoir. Intergrown plate-shaped crystals appeared after 1 day and grew to full size over 3 days. Crystals were cryoprotected in reservoir solution doped with 20% di-ethylene glycol prior to flash-cooling in liquid nitrogen. 360° diffraction data were collected at 100 K on beamline MX1 of the Australia Synchrotron (Melbourne, Australia) and processed with the XDS package[41,42]. Crystals belong to space group C2221 and contain one copy of the nanoCLAMP8-VHH fusion protein in the asymmetric unit. The structure was solved by molecular replacement in PHASER[43], using coordinates of VHH (PDB:3QXV) and nanoCLAMP (PDB:2W1Q). The structure was refined to 1.83 Å resolution in REFMAC5[44] and COOT[45]. The MTX ligand was added in the final rounds of refinement, using a restraints library generated with PRODRG[46]. The quality of the final electron density map together with a comparison of the refined temperature factors of the ligand and the surrounding protein residues suggest that the binding site is fully occupied by MTX (Fig. 3e). Full data collection and refinement statistics are summarized in Table 1. Structure factors and coordinates have been deposited in the Protein Data Bank with accession code 7RG7. The crystal of nanoCLAMP3:VHH:MTX was obtained by mixing 0.2 μl of the protein–MTX complex solution in 20 mM Tris–HCl pH 7.4, 40 mM NaCl, 5 mM DTT at 20 mg/ml and 0.2 μl of reservoir solution containing 1.4 M sodium malonate–malonic acid pH 7.0. The drops were set up in a sitting drop plate (SD2, SwissSci, UK) and incubated at 20 °C against 50 μl reservoir. Crystals appeared after 2 days. Cryoprotection was achieved by addition of reservoir solution doped with 20% glycerol prior to flash-cooling in liquid nitrogen. 360° diffraction data were collected at 100 K on beamline MX2 of the Australia Synchrotron (Melbourne, Australia) for each crystal and processed with the XDS package. Two data sets were combined to obtain the final data set for structure solution. Crystals belong to space group I2 and contain seven copies of the nanoCLAMP3-VHH fusion protein in the asymmetric unit. The structure was solved by molecular replacement in PHASER, using coordinates of VHH (PDB:3QXV) and nanoCLAMP (PDB:2W1Q). The structure was refined to 2.9 Å resolution in REFMAC5 and COOT release 0.9.4.1. The MTX ligand was added in the final rounds of refinement, using a restraints library generated with PRODRG. The quality of the final electron density map together with a comparison of the refined temperature factors of the ligand and the surrounding protein residues suggest that the binding site is fully occupied by MTX (Supplementary Fig. 5C). Full data collection and refinement statistics are summarized in Table 1. Structure factors and coordinates have been deposited in the Protein Data Bank with accession code 7RGA. Structures were analyzed and figures were prepared using ICM version 3.9-1b (Molsoft).

**HDX-MS analysis of MTX-dependent interaction of nanoCLAMP5 with VHH**. To identify MTX-dependent binding interactions between VHH:MTX in complex with nanoCLAMP5 we performed comparative HDX-MS analyses in the presence or absence of methotrexate. A LEAP HDX-2 Automation manager was used to automate labeling, quenching and injection of samples into a ACQUITY UPLC M-Class HDX Manager (Waters, Milford, MA, USA). 3 µl Purified protein (30 µM in 20 mM Tris/HCl pH 7.2, 100 mM NaCl) was incubated in 57 µl PBS buffer reconstituted in $D_2O$ (99.90%, Sigma). VHH:NC5 and MTX:VHH:NC5 complexes were prepared at a molar ratio of 1:1: and 3:1:1, and pre-incubated for 30 min to achieve binding prior to each hydrogen–deuterium exchange reaction. Deuterium labeling was performed for 30 s, 1, 10, 100 and 200 min, followed by quenching of 50 µl of the deuterium exchange reaction mixture in 50 µl pre-chilled 50 mM PBS quench solution to lower the pH to 2.5 and lower temperature to 0 °C. 80 µl Quenched samples were injected onto chilled ACQUITY UPLC M-Class HDX Manager (Waters, Milford, MA, USA). Samples were subjected to online digestion using immobilized Waters Enzymate BEH pepsin column (2.1 × 30 mm) in 0.1% formic acid in water at 100 µl/min. The proteolyzed peptides were trapped in a 2.1 × 5 mm C18 trap (ACQUITY BEH C18 VanGuard Pre-column, 1.7 µm, Waters, Milford, MA, USA). The proteolyzed peptides were eluted using acetonitrile and 0.1% formic acid gradient (5–40% 7 min, 40–95% 1 min, 95% 2 min) at a flow rate of 40 µl/min using an ACQUITY UPLC BEH C18 Column (1.0 × 100 mm, 1.7 µm, Waters, Milford, MA, USA) pumped by UPLC M-Class Binary Solvent Manager (Waters, Milford, MA, USA). Positive electrospray ionization source fitted with a low flow probe was used to ionize peptides sprayed onto SYNAPT G2-Si mass spectrometer (Waters, Milford, MA, USA). Data was acquired in $MS^E$ acquisition mode using 200 pg/µl Leucine enkephalin and 100 fmol/µl [Glu1]-fibrinopeptide B ([Glu1]-Fib) at a flow rate of 5 µl/min for lockspray correction.

ProteinLynx Global Server (PLGS) v3.0 was used to identify peptides from non-deuterated protein samples. The identified peptides were further filtered in DynamX v3.0 using a minimum intensity cutoff of 5000 for product and precursor ions, minimum products per amino acids of 0.3, a precursor ion mass tolerance of <5 ppm and a file threshold of 3, using DynamX v.3.0 (Waters, Milford, MA, USA). Deuterium exchange plots, relative deuterium exchange and difference plots were generated. All deuterium exchange experiments were performed in triplicate and reported values were not corrected for deuterium back exchange. HDX-MS data were analyzed using the HD-eXplosion software tool and visualization of hydrogen–deuterium exchange data with statistical filtering[47].

**MST analysis of binding VHH and nanoCLAMP in the absence and presence of MTX**. To determine the $K_d$ between nanoCLAMP8 and VHH in the presence and in the absence of MTX MicroScale Thermophoresis (MST) experiment were performed using Monolith NT.115 GREEN/RED (NanoTemper Technologies GmbH, Germany). Monolith NT.115 Standard Treated Capillaries and buffer containing 20 mM Tris–HCl pH 7.2, 100 mM NaCl were used for all titration experiments. Firstly, we titrated 10 nM GFP-VHH with an increasing concentration of MTX (LED power 90%, MST power 80%) to obtain the $K_d$ value 4.3 ± 0.25 nM, which was close to the published value[22] (Supplementary Fig. 7). Further, we titrated 500 nM GFP-VHH with increasing concentration of nanoCLAMP8 (LED power 20%, MST power 20%) to obtain the $K_d$ value 3.7 ± 0.23 µM. Higher concentrations of the reporter were used to obtain a more stable signal. Finally, we titrated 10 nM GFP-VHH with increasing concentration of nanoCLAMP8 in the presence of 10 µM MTX (LED power 90%, MST power 80%) to obtain the $K_d$ value 8.2 ± 0.43 nM. The fit of the data was carried out using MO.Control software.

**Analysis of MTX concentration in patient serum using an assay based on GDH-based-MTX biosensor**. The assays were performed in 1 ml reaction volume containing 0.25, 2.5, or 25 µl serum sample, 60 µM electron accepting dye dichlorophenolindophenol in the presence of 0.6 mM electron mediator phenazine methosulphate, 1 mM $CaCl_2$, 10 nM of GDH-CaM-VHH fusion and 100 nM of nanoCLAMP5-CaM-BP in the assay buffer containing 20 mM Tris, pH 7.2, 100 mM NaCl. The mixture was incubated at 25 °C for 30 min without glucose. Upon addition of glucose to the final concentration of 20 mM the change of 600 nm absorption was monitored. The observed reaction rates of individual experiments were obtained by linear fitting of absorption changes during first 4 min. The $k_{obs}$ then was used to determine MTX concentration by correlating it with the calibration curve generated by performing the assay with known concentrations of MTX (Supplementary Fig. 8). The calibration curve was obtained by titration of the sensors with the known amount of MTX (Sigma-Aldrich) and the data was analyzed using Prism 9.2 software.

**Analysis of MTX concentration in patient serum using immunochemistry station**. Serum methotrexate was measured with the Architect Methotrexate assay on an Architect i2000 SR analyzer (Abbott, Lake Bluff, IL, USA) and according to the instructions of the manufacture. The assay is a chemiluminescent microparticle immunoassay for the quantitative determination of methotrexate in human serum and plasma.

**Solution electrochemical measurements**. For solution electrochemistry assays 20 µl reactions containing 0.5 µM VHH-GDH-CaM and 1.5 µM nanoCLAMP5-CaM-BP fusion were pre-incubated for 20 min ± 2 µM MTX, 1 mM $CaCl_2$ and 20 mM NaCl in 20 mM Tris buffer (pH 7.2) and 2.5 mM mPMS. The electrochemical measurements were performed using Digy-Ivy potentiostat operated by DY2116B software. To start the electrochemical reaction 20 mM glucose were added and 10 µl of sample were transferred to DropSens Gold screen printed electrode with coverslip placed over the droplet to reduce atmospheric oxygen transfer and interference with the reaction. The working electrode was polarized for 0.5 s at +0.1 V vs. Ag reference strip at 0.3 and 6 min, with a signal (nA) reading at 0.5 s used for calculation each polarization cycle. Sensor enzymatic activity was reported as maximum nA increasing per minute, due to reduced mPMS accumulating in the reaction (Supplementary Fig. 10A).

**Thermostability analysis of the GDH-based biosensor**. Thermostability of PQQ-GDH and related oxidoreductases is one of the important features that enables rapid and large-scale production and long-term storage of disposable glucose sensors. We, therefore, decided to test the thermostability of the developed biosensor and its components to assess their compatibility with manufacturing processes that include high temperature drying of the protein component on the electrode surface. To this end we incubated VHH-GDH-CaM, nanoCLAMP5-CaM-BP as well as the parental GDH-CaM switch module in triplicates at different temperatures for indicated periods of time and then performed their activity assay. Activity of GDH-CaM was induced by addition excess of M13 CaM-BP. VHH-GDH-CaM was activated either by addition of M13 CaM-BP or by addition of fresh nanoCLAMP5-CaM-BP in the presence of excess of MTX. Activity of nanoCLAMP5-CaM-BP was tested by the addition of fresh VHH-GDH-CaM and excess of MTX. Activity of CaM-GDH was tested by addition of M13 CaM-BP. The incubation times were as follows: 4 °C for 3 h, 25 °C for 3 h, 37 °C for 3 h, 50 °C for 30 min, 50 °C for 10 min, 80 °C for 10 min. All reactions were incubated with their cognate ligands for 30 min prior to the addition of the electron mediator, the dye and glucose.

**Construction and characterization of methotrexate and rapamycin sensory bioelectrodes**. Bioelectrodes were prepared using highly porous gold-modified 4 mm screen-printed gold electrodes (*DRP-220AT*, Metrohm DropSens) using cysteamine/glutaraldehyde method described previously with minor modifications (Supplementary Fig. 10B)[32].

To produce the bio-electrodes that can respond reversibly to the changes in the analyte concentration we modified the GDH-CaM two-component biosensor system reported previously by mutating two cysteines at the linker region of inserted CaM to serine thereby preventing formation of a disulfide bond with cysteine in the insertion position that is expected to lock the biosensor in its activated conformation[23] (Table S1). Interestingly, the mutant protein showed improved dynamic range compared to its parental protein indicating the need for more detailed analysis of the linker role in biosensor's function[48,49].

Electrochemical measurements were carried out in 2 ml miniaturized electrochemical cell (Supplementary Fig. 10 using cyclic voltamperometry. Initially base voltamperogram was collected with 2 mV/s scan rate in buffer solution (25 mM Tris–$H_2SO_4$ buffer, pH 7.2, 100 mM $Na_2SO_4$ and 1 mM $Ca(CH_3COO)_2$). Subsequently 20 mM glucose was added, mixed by pipetting and a cyclic voltamperometry scan at the same scan rate was repeated. The third scan was carried out after addition of 200 nM Binder-CaM-BP fusion to the previous solution. The last scan was performed after addition to the solution of the ligand to a final concentration of 200 nM followed by 30-min incubation at room temperature and voltametric scanning (Fig. 5d). The data represents technical replicates performed three times.

To assess the reversibility of the electrode activation the bioelectrodes these were extensively washed twice for 10 min in 3 ml buffer solution using shaker for agitation. The reversal of the activation was not complete as approximately twice higher basic signal comparing with the first one (6.2–7.1 µA vs. 3.8 µA at potential +0.4 V vs. Ag/AgCl/3 M KCl) was observed (Fig. 5e upper inset). It could be explained by presence of residual amount of the activating component tightly attached to the immobilized GDH component of the biosensor.

**Reporting summary**. Further information on research design is available in the Nature Research Reporting Summary linked to this article.

## Data availability

The authors declare that the data supporting the findings of this study are available in this paper and its supplementary information files. Source data are provided as a Source Data file. The structure factors and coordinates used in this study are available in the Protein Data Bank under accession codes 7RG7 and 7RGA. Source data are provided with this paper.

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

## Acknowledgements

We thank Dr. Carel Pretorius from Pathology Queensland for carrying out quantification of biomarkers in human samples for this study. Authors are thankful to Brett Collins for his help with the MST experiments. We would also like to thank the Melbourne Mass Spectrometry and Proteomics Facility of The Bio21 Molecular Science and Biotechnology Institute at the University of Melbourne for the support of mass spectrometry analysis. We thank the Australian Synchrotron for beamtime and the beamline scientists for their help with data collection. This work was supported in part by the Australian Research Council Discovery Projects DP160100973, DP150100936 as well as ARC Centre of Excellence in Synthetic Biology CE200100029 to K.A. The work was also supported by NHMRC Development grants APP1113262 to K.A. and APP1179001 to K.A. and J.P.J.U. This work was also in part supported by HFSP grant RGP0002/2018 to K.A. and E.K. and US Department of Defense grant W81XWH-20-1-0708 to K.A., E.K., and A.M. K.A. gratefully acknowledges financial support of CSIRO-QUT Synthetic Biology Alliance and QUT Industry Engagement Fund.

## Author contributions

Z.G.—designed experiments, performed experiments, analyzed data and wrote manuscript, O.S.—designed experiments, performed experiments, analyzed data and wrote manuscript, W.A.J.—designed experiments, performed experiments, analyzed data and wrote manuscript, P.W.—performed experiments, J.P.J.U.—designed experiments and analyzed data, T.S.P.—designed experiments, performed experiments, analyzed data and wrote manuscript, J.N.—designed experiments, performed experiments, analyzed data, J.P.—analyzed data, T.N.—performed experiments, J.P.J.U.—designed experiments and analyzed data, C.H.—designed experiments, performed experiments, analyzed data, A.M.—designed experiments, analyzed data, R.J.S.—designed experiments, performed experiments, analyzed data and wrote manuscript, E.K.—designed experiments, analyzed

data and wrote manuscript, K.A.—designed experiments, analyzed data and wrote manuscript.

## Competing interests

The authors declare the following competing interests: Z.G. and K.A. are named inventors on patents covering electrochemical protein biosensor technology used in this study. K.A. holds equity in Molecular Warehouse Ltd that owns one of those patents. The rest of the authors declare no competing interests. RJS is a named inventor on a patent covering nanoCLAMP technology, and holds equity in Nectagen, Inc.
