## [Peer Review File · Nature Communications]

Reviewers' Comments:

Reviewer #1:

Remarks to the Author:

In NCOMMS-21-13428-T, Guo et al. develop a chemically-induced protein dimerization system with the goal of using it as a platform for measuring levels of therapeutic drugs at point-of-care or by continuous monitoring. Starting with three methotrexate (MTX) binding proteins (DHFR, thymidylate synthase, and a camelid VHH domain previously engineered to recognize MTX), the authors screen a phage display library of nanoCLAMP variants for those that bind to the protein-MTX complexes but not to the proteins alone. They obtain several VHH variants that exhibit MTX-dependent binding. MTX binding is reported by fusing the VHH and nanoCLAMP domains to CaM-GDH and CaM peptide, as described in the author's previous work. Proof-of-concept is demonstrated by colorimetric and electrochemical detection of MTX in human serum and buffer.

Except for the technical issues mentioned at the end, the experiments are well designed with appropriate controls. The main conclusions are generally well supported. For cancer patients undergoing high dose MTX treatment, serum drug concentration is periodically measured to ensure that it does not approach toxic levels, and the authors present very nice data showing that their sensor gives colorimetric results comparable to those of a commercial MTX diagnostic kit. It is not discussed, however, whether the present technology offers any advantages over existing methods. It's also unclear if continuous MTX monitoring is commonly done or is needed, and whether the electrode can work with real-world, dirty samples.

From a protein engineering standpoint, the novelty and potential generalizability of the sensor design appear to be somewhat limited. The MTX recognition domain was generated by biopanning available libraries, and the enzymatic/potentiometric output system has been described in several papers. Broad-based appeal to readers of Nature Communications could come from the potential generalizability of the sensor design. However, I find the assertions of generality to be overstated. Success depends on first identifying a suitable "anchor" protein that binds the drug with high affinity, and then finding a nanoCLAMP variant that exhibits drug-dependent binding over a concentration range commensurate with fluctuations in therapeutic drug levels. Thus, future designs seem conditional on largely unknown factors, most notably the success of library screening steps. Only one of three anchor proteins worked in the present case, and it's interesting that it proved to be an engineered nanobody rather than one of the natural targets of the drug (the latter being generally more available as anchors compared to the former).

The authors should include a more detailed description of the statistical methods. How are replicates defined (technical vs biological/experimental, and how many of each were performed)? Are uncertainties expressed as standard deviation, standard error, etc.? Error bars are missing from many of the figures. Raw data should be included whenever possible, e.g. Fig. 4A. There are multiple typographical errors and mis-referenced figures. The usage of "thereon" in the title and text is a bit confusing.

Reviewer #2:

Remarks to the Author:

The authors demonstrate a novel ternary complex approach for engineering binding-linked conformational changes/binding-linked oligomerization into two small-molecule-drug-binding proteins, with the goal being to couple recognition of that small molecule with an output signal. Focusing primarily on MTX and less so on rampamycin, they first used phage display to select for proteins that would form ternary complexes with the analyte and one or more of its naturally occurring binding proteins. Upon success in this endeavor, they engineered large molecular complexes comprised of two parts. One part included the newly selected ternary component and a peptide that binds calmodulin with low affinity. The other part included the target receptor coupled to a glucose dehydrogenase split with a calmodulin domain such that it exhibits no significant enzymatic activity unless the calmodulin domain binds to its target protein, causing a conformational change that leads to activity. Adding these two components together the authors reported, as they expected, target-induced ternary complex formation that lead to activation of

the GDH and subsequent detection of its product as a means of quantifying the target.

This is nice work, carefully done. The selection for ternary complex forming proteins is impressive (even if, as noted by the authors, is not entirely new). The coupling with calmodulin-modulated GDH is clever. Getting all of the pieces to "play well together" and work in concert is impressive. This work is rather a "tour de force." None of the pieces is 100% novel, but in getting them to all work together so nicely, the authors have really shown what can be done by dedication and work.

My quibbles are quite minor. The authors note that "Toxicity of [MTX] requires its monitoring in patient's serum and therefore a MTX Point-of-Care or continuous monitoring system would have a significant clinical value." My two quibbles are: (1) TDM is not the standard of care for MTX. Perhaps it should be, but it is not. Given that "requires" is far too strong a word here. Is recommended? Would improve outcomes? Those work. (2) the use of "continuous" monitoring in this sentence lead me to believe that the authors might have an approach towards continuous monitoring. They do not: they have to add a secondary reagent (the ternary-forming protein/calmodulin peptide); unless they can supply that continuously, they can't do continuous monitoring. And the only place I can see where continuous monitoring would be of value is in the body, where they cannot provide that reagent.

A second, perhaps related quibble. The sensor's out put will depend on the concentration of glucose and oxygen, and not just on the concentration of the target. This, too, would likely preclude its use in continuous monitoring in the body. Perhaps the latter thing is a quibble, but at least a small mention of the O₂ and glucose dependence of the output and how these can be dealt with is in order.

Reviewer #3:

Remarks to the Author:

The manuscript by Alexandrov and coworkers reports on the development a methotrexate responsive protein dimerization system and its subsequent integration into a previously developed bioelectronic assay system based on allosterically regulated PQQ-glucose dehydrogenase (PQQ-GDH). The work is presented as a generic new method to evolve chemically induced dimerization (CID), but in this paper the engineering is confined to methotrexate as the analyte of interest. Three different structurally characterized methotrexate-protein complexes are used as starting points (anchors), dihydrofolate reductase, thymidylate synthase and an MTX-binding VHH nanobody. Using phage display, a library of nanoclamp domains is screened for MTX-dependent binders. After 3 rounds of selection and amplification, MTX-dependent binders are only found for the MTX-VHH complex, but not for the two other MTX-binding proteins (actually one clone is reported for DHFR, but this one is not pursued further). Structural characterization of two of the binding clones reveals the that NanoClamp proteins interact with the MTX-VHH complex via two types of interactions. Surprisingly, the variable part of the NanoClamp protein interacts with a part of the VHH domain that does not undergo a structural change upon MTX binding. The second interaction occurs between a constant part of the NanoClamp protein and a part of the VHH domain that gets exposed upon MTX binding. The latter explains why NanoClamp binding was only successful for the VHH-MTX complex and not the other receptors. As such one could argue whether the proposed strategy is generic, although it might be interesting to see whether it also applies to other VHH-ligand interactions. The best performing Nanoclamp variant shows an impressive and very useful difference (1000-fold) in affinity between the MTX-VHH complex and VHH alone, which is subsequently used to control the activity of the PQQ-GDH system. Overall, the work represents an interesting example of the generalizable protein biosensor concept previously reported by the Alexandrov group (ref 22, 24) for a biomedically relevant drug. The work is complete in that it reports extensive protein engineering to develop ligand responsive protein-protein interaction, their X-ray structural characterization and a demonstration of the system for bioelectronic detection. I therefore support the publication of the work provided the questions and comments listed below are appropriately addressed.

Questions and comments:

- The Johnsson group also developed a family of methotrexate responsive bioluminescent protein

switches using their LUCID system. Their system is also based on DHFR but uses a competition mechanism. It would be useful and appropriate for the authors to compare their sensor to this previously developed system that also allows POC detection of methotrexate.

- The NanoClamp system is not a widely known alternative antibody system, please provide a little bit more background (what protein core is it based on, what has it been used for, did the authors chose this system instead of other system?)

- Figure 2D. Please align the + and - better on top of the gels to the different lanes

- Figure 4A/B. Why was a much higher concentration of EGFP-VHH (500 nM) used in the nanoCLAMP titration experiment in the absence of MTX than in its presence (10 nM). Please note that it would be impossible to detect a low K_d affinity using 500 nM of EGFP-VHH. In other words, can the authors show that the low microM K_d is not an apparent K_d that results from using a high concentration of EGFP-VHH? Most straightforward would be to repeat this titration with 10 nM EGFP-VHH as well.

- I am a little confused by the statement that the authors can measure low nM concentrations of MTX in patient serum samples. In the SI it is mentioned that serum samples are diluted 40-4000-fold before they are assayed, so the authors cannot claim that they can do these assays in serum. Also, do the concentrations refer to the concentration in serum before dilution or the final concentration in the assay mixture?

- Experiments with patient serum samples were done using optical detection, which may explain the need for dilution. Did the authors test whether their final bioelectronic assay could be used with (diluted) serum samples?

- "The combination of high specificity with nanomolar sensitivity makes the proposed technology a potential contender to the leading methods of continuous biochemical monitoring technologies." -> I think this not correct for two reasons. First, the assay still need reagents, which makes it challenging to be used for continuous monitoring. Second, the performance of the sensor in complex medium such as undiluted serum has not be shown.

Reviewer #4:

Remarks to the Author:

The authors presented successful development of MTX biosensor by utilizing anti-MTX VHH and binders (nanoCLAMP) that interact strongly with VHH in the presence of MTX. The developed biosensor could detect MTX in solution with high sensitivity such as nM concentration of MTX and also could be used for electrochemical sensor by fusing with other protein components. The experimental design and results are quite high quality.

However four concerns were raised as follows;

1. This work is an example of development of MTX sensor, while I suspect whether the demonstrated approach in this manuscript can be used for other small molecules than MTX. The author set title of the manuscript as generalized one as if the approach, CID, shown here can be applied for other small molecules.

2. The variable elements bind to beta-sheet of VHH and loop 64-71 binds to CDR1 and CDR4 of VHH and this binding moiety seems not as allosteric binding because conformational arrangements of CDRs 1 and 4 which are binding sites for the binder are induced by the recognition of MTX. Therefore molecular scheme depicted in figures including Fig4 seems to be misleading whereas I can agree with the authors' statement "the ligand induced conformational change creates an additional binding epitope for the nanoCLAMP" (P10 the bottom).

3. Structural analysis of nanoCLAMP-GDH+MTX system was not shown which is necessary for supporting whether the outcome of the panning meets with the authors' idea including allosteric binding.

4. In fig.4C it looks that VHH was fused to GDH-CaM chimera and nanoCLAMP was fused to CaM-BP, while figure caption of fig4D explains that VHH-CaM-GDH and nanoCLAMP- CaMBP were used. This is quite confusing and difficult to understand completely.

Minor concerns;

5. P4L10 why did the authors specify Fig1B here?

6. P7L19 "It is clear that the rearrangement of the CDR1 region (Ser199-W206) of VHH upon MTX binding plays a key role in selective binding of the nanoCLAMP." was not evident from Fig3C rather its seems that only slight change occurred.

7. P7L15 Although the authors stated "This arrangement is quite surprising as one would expect the selection to result in structures where the variable elements would interact with the ligand or its surrounding.", I do not have same impression since the authors did not explain the intended structural basis for the binder recognition of target (MTX-VHH complex in this case).

8. Calcium ion is found in the binder structure. however how did the authors control the involvement of calcium ion? Were there possibility of failure in getting positive clones because of lack in the calcium ion during the panning?

9. Typos were found like P9L1 "revieald", need to be corrected.

10. P13 Fig. 9B, C should be Figs. S9B, C

We thank all reviewers for their positive and constructive comments, and for recognising the value and potential impact of this study. We have introduced a number of modifications into the manuscript according to reviewer suggestions. In addition, we now provide data on the performance of MTX bioelectrodes in human serum (Fig.4F and S10E). We have also added additional data on the analysis of thermostability of the biosensor component to test their compatibility with typical test strip manufacturing processes (Fig.S9 and the associated text).

We addressed specific comments of the reviewers in the following way:

Reviewer #1

Point 1. For cancer patients undergoing high dose MTX treatment, serum drug concentration is periodically measured to ensure that it does not approach toxic levels, and the authors present very nice data showing that their sensor gives colorimetric results comparable to those of a commercial MTX diagnostic kit. It is not discussed, however, whether the present technology offers any advantages over existing methods.

Response: There are two main clinical assays used for methotrexate qualification. A) a turbidity assay based on aggregation of latex beads functionalised with methotrexate antibody. This assay is rapid and can be run on simple clinical chemistry workstations, but is only suitable for measuring high methotrexate concentrations typically present in samples of cancer patients B) Immunochemical assays that are run on more sophisticated immunochemistry stations that can detect low nanomolar concentrations of methotrexate present in the samples of patients where the drug is used as an immunosuppressive agent (i.e. arthritis and Crohn's disease).

The presented protein biosensor-based solution assay can quantify low nanomolar concentration of methotrexate and can be performed using plate readers or open clinical chemistry workstations combining assay simplicity with high sensitivity.

We included the following phrase on the page 12 of the manuscript to reflect the arguments above:

The sensitivity and simplicity of the assay allows transfer of the assay from complex immunochemistry stations onto more simple and ubiquitous clinical chemistry analyzers and plate readers. Such tests can then be performed in smaller diagnostic laboratories and community clinics, thus improving patient access to MTX monitoring.

Point 2 It's also unclear if continuous MTX monitoring is commonly done or is needed, and whether the electrode can work with real-world, dirty samples.

Response: We have rewritten the paragraph on page 14 discussing the utility of the developed biosensors in end point and continuous monitoring of MTX.

We alert the reader to the fact that the assay can be ported onto dry electrochemical strips that could enable Point-of-Care methotrexate monitoring in patients with inflammatory conditions such as arthritis and Crohn's disease, where long-term drug application requires careful dosing and adherence monitoring. We provide a reference to the sources discussing the utility of personalised MTX therapy.

Continuous monitoring of methotrexate would be of value in cancer treatment where the drug can lead to renal failure and systemic toxicity. Monitoring the kinetics of drug clearance would provide an early warning of potential renal failure and ensuing toxic effects. To summarise these arguments, we have now written the following paragraph on page 14:

The combination of high specificity with nanomolar sensitivity makes such electrodes an excellent starting point for developing Point-of-Care MTX testing systems. Such systems could enable monitoring of adherence as well as MTX therapy personalization in chronic inflammatory conditions such as arthritis and Crohn's disease³². Furthermore, the proposed technology is a potential contender to become a leading method in continuous biochemical monitoring. In application to methotrexate, it may be used to monitor drug clearance and help to mitigate renal and systemic toxicity.

We also now include results of the experiments where we tested performance of the MTX electrodes in 50% human serum. These experiments are now described on page 14.

We finally tested the ability of the developed electrode system to respond to their cognate ligand in human samples. To this end we repeated the experiments using MTX electrode and 50% human serum spiked with different concentrations of the drug. The electrode retained its functionality, although with higher background signal (Fig.5H and S10E,F). Further work will be required to optimize the electrochemical parameters of the assay to reduce the background signal as well as find ways of tackling the high viscosity of the sample (which complicates work with undiluted serum samples).

Point 3 *From a protein engineering standpoint, the novelty and potential generalizability of the sensor design appear to be somewhat limited. The MTX recognition domain was generated by biopanning available libraries, and the enzymatic/potentiometric output system has been described in several papers. Broad-based appeal to readers of Nature Communications could come from the potential generalizability of the sensor design. However, I find the assertions of generality to be overstated. Success depends on first identifying a suitable "anchor" protein that binds the drug with high affinity, and then finding a nanoCLAMP variant that exhibits drug-dependent binding over a concentration range commensurate with fluctuations in therapeutic drug levels. Thus, future designs seem conditional on largely unknown factors, most notably the success of library screening steps.*

We agree with the referee that construction of chemically induced dimerization systems (CIDs) and their use in biosensor construction is not trivial. However, over the last three years there has been a steady stream of reports on computational and experimental approaches for CID design. While the reviewer correctly suggest that in our approach we relied on the existence of an MTX-binding anchor domain, it is worth noting that the VHH:MTX complex is itself created through phage selection of immune MTX VHH library. A recent study (reference 13) demonstrated that VHH-naive library could be used to select both the anchor as well as the second "lid" domain of CID. Further, a lot of progress has been made in developing chemically regulated transcription factors that can be utilised as anchor domains in CID design efforts. While all these approaches are far from simple, the improvements in selection systems, design of the binder libraries, and of the selection campaigns improve the odds of successful designs of CIDs.

We now added discussion to the page 16 of the manuscript, in order to better discuss these points.

Furthermore, in silico design coupled to diversity-based screening was recently successfully used to construct a CID system regulated by farnesyl pyrophosphate, further expanding the number of avenues available for de novo construction of such systems¹¹. The sophistication of the methods used still presents a challenge for their broad adoption, but constant methodological progress promises to address this limitation.

Point 4 *Only one of three anchor proteins worked in the present case, and it's interesting that it proved to be an engineered nanobody rather than one of the natural targets of the drug (the latter being generally more available as anchors compared to the former).*

Response: We actually found MTX-dependent binders to two of the three anchor proteins: 4 unique B-DHFR/MTX and 9 unique B-VHH/MTX candidate nanoCLAMPs, each from a pool of 95 random clones (from panning round 3). A higher throughput screen (many labs look at thousands at a time) would likely improve odds of success. The failed thymidylate synthase anchor also has a lower affinity for MTX. As complex formation is a requirement for binder selection, it is likely that not enough thymidylate synthase /MTX complex was formed for phage enrichment, suggesting that high affinity ligand:anchor pairs are more suitable for this technique. We have now rewritten a part of the discussion section in the following way:

This mechanism helps to rationalize the differences in the outcomes in binder selection against different MTX:protein complexes. VHH displays the largest localized conformational change in its CDR1 loop upon ligand binding. It is followed by dihydrofolate reductase where MTX binding leads to a small change in the position of the α -helix over the binding site. In this case fewer ligand-dependent nanoCLAMPs were found and the difference in interaction strength with and without ligand was lower (Fig.S3). The smallest conformation change upon MTX binding is detected in thymidylate synthase where no MTX-dependent binders were identified (Fig. S10). Thymidylate synthase complex with MTX has the lowest affinity among complexes tested, which may have adversely affected the selection process³⁶. However, we sampled only a relatively small number of clones and further screening may have resulted in identification of nanoCLAMPs with better selectivity for DHFR:MTX complex.

Point 5 *The authors should include a more detailed description of the statistical methods. How are replicates defined (technical vs biological/experimental, and how many of each were performed)? Are uncertainties expressed as standard deviation, standard error, etc.? Error bars are missing from many of the figures. Raw data should be included whenever possible, e.g. Fig. 4A.*

On request of the referee, we added standard deviations to the K_d values obtained from the fit of the data to quadratic or linear equations as listed material and method sections. We now provide the raw data of titration experiments as tables S2 and S3 that also include the standard deviation of the data fits. We could not add error bars to the individual data points on some graphs as due to the very low SD the bars would be smaller than the dots. We also indicate the number of experimental replicates in corresponding methods sections and, in some cases, in the figure legends.

Point 5 *There are multiple typographical errors and mis-referenced figures. The usage of "thereon" in the title and text is a bit confusing.*

The manuscript was carefully proofread for correct figure referencing as well as typographical errors. On the suggestion of the referee, we changed the title to "Design of chemical dimerization systems and their use in bio-electronic devices".

Reviewer #2

Point 1 *The authors note that "Toxicity of [MTX] requires its monitoring in patient's serum and therefore a MTX Point-of-Care or continuous monitoring system would have a significant clinical value." My two quibbles are: (1) TDM is not the standard of care for MTX. Perhaps it should be, but it is not. Given that "requires" is far too strong a word here. Is recommended? Would improve outcomes? Those work.*

Response 1: We thank the reviewer for the positive comment on our study but must politely disagree with the statement that therapeutic monitoring of methotrexate is not common. If this was the case, we would not be able to obtain patient samples with reference data from the collaborating chemical

pathology laboratory. The standard clinical assay used for this purpose is described on the page 15 of Supplementary materials section.

We agree, however, that we have not fully explained the utility of the methotrexate monitoring for relevant clinical conditions. We discuss this point more explicitly now (see response to the reviewer 1, points 2 and 3).

Point 2 *The use of "continuous" monitoring in this sentence lead me to believe that the authors might have an approach towards continuous monitoring. They do not: they have to add a secondary reagent (the ternary-forming protein/calmodulin peptide); unless they can supply that continuously, they can't do continuous monitoring. And the only place I can see where continuous monitoring would be of value is in the body, where they cannot provide that reagent. A second, perhaps related quibble. The sensor's output will depend on the concentration of glucose and oxygen, and not just on the concentration of the target. This, too, would likely preclude its use in continuous monitoring in the body. Perhaps the latter thing is a quibble, but at least a small mention of the O₂ and glucose dependence of the output and how these can be dealt with is in order.*

This is a valid point, but it creates an unsurmountable obstacle only in the case of protein analytes. In case of small molecules, the electrodes can be separated from the medium with a semi-permeable membrane that maintains a constant concentration of the soluble biosensor component at the electrode. Use of such membranes is common in modern electrochemical biosensors and is now referenced in the paper. The issue of fluctuations in blood glucose concentration is indeed pertinent but can be addressed by either reducing the K_m of the protein biosensor for glucose or by introducing a reference glucose electrode. As we use oxygen-independent enzyme, fluctuations in the oxygen do not affect the performance of the biosensor. We added the following paragraph to the page 17 of the manuscript to discuss these issues.

One can also envisage that a similar approach can be utilized for construction of small molecule continuous monitoring systems that combine electrode immobilized GDH-CaM-Binder1 unit with soluble Binder2-CaM-BP. Such a system can be separated from the sample with a semi-permeable membrane, thus ensuring that the soluble component does not diffuse away and can associate with the electrode-bound reporter in a ligand-dependent manner. Use of such semi-permeable membranes in electrochemical biosensors is well established and would enable rapid transport of the analyte and circulating glucose to the biosensor⁴⁰. Although fluctuations of glucose levels will influence the accuracy of this approach, it can be overcome by either reducing the K_m of the GDH or by including a reference glucose electrode into the assembly.

Reviewer #3

Point 1 *The Johnsson group also developed a family of methotrexate responsive bioluminescent protein switches using their LUCID system. Their system is also based on DHFR but uses a competition mechanism. It would be useful and appropriate for the authors to compare their sensor to this previously developed system that also allows POC detection of methotrexate.*

Response: The reported apparent IC₅₀ (inverse of apparent K_d) of the MTX LUCID biosensor was reported to be 750nM with a detection limit of 100 nM. Given that NanoLuc-based protein biosensors do not perform well in serum concentrations above 10%, the LUCID biosensor cannot be used for management of inflammatory conditions, and has utility only for monitoring MTX in cancer chemotherapy. Another complexity of using NanoLuc as a biosensor component is its substrate furimazine, which is both unstable and proprietary thus complicating its use. The GDH-based MTX biosensor displayed a limit of detection 100 times lower than LUCID and utilises glucose and cheap

electron mediators. Finally, the construction of LUCID required protein semi-synthesis which is significantly more complicated than recombinant production of GDH-based biosensors in *E.coli*.

The list of differentiating features can go on, but to avoid distracting the reader from the central points of the paper we summarise our response in the following sentence now in page 10:

The observed limit of detection was below 1nM signifying an over 100-fold sensitivity improvement over the previously reported bioluminescent MTX biosensor²⁷.

Point 2 *The NanoClamp system is not a widely known alternative antibody system, please provide a little bit more background (what protein core is it based on, what has it been used for, did the authors chose this system instead of other system?)*

Response: nanoCLAMPs (nano-Clostridial Antibody Mimetic Proteins) are based on an immunoglobulin-like, thermostable carbohydrate binding module from a *Clostridium hyaluronidase* that are small 15 kD proteins devoid of cysteines that are easily expressed and purified from the cytosol of *E.coli*. They can be denatured and refolded without loss of activity, and are amenable to fusions at both N and C termini, which are adjacent to each other for modular engineering. Low nanomolar binders can typically be selected from large naive phage display libraries to diverse targets. Finally, nanoCLAMP targets can be released by exposure to polyol and chaotropic salt at neutral pH due to their polyol responsive nature, which could enable sensor reuse.

In order to keep the main text of the manuscript concise, we added the following description to page 5:

We chose the nanoCLAMP domain due to its small size (15kDa), ease of recombinant production, and finally the lack of cysteine residues enabling its use in both oxidizing and reducing environments. The ability to reversibly disrupt nanoCLAMP:ligand interaction with polyol and chaotropic salt was also seen as a potentially beneficial feature¹⁸.

Point 3 *Figure 2D. Please align the + and – better on top of the gels to the different lanes*

Response: The markers are now properly aligned.

Point 4 *Figure 4A/B. Why was a much higher concentration of EGFP-VHH (500 nM) used in the nanoCLAMP titration experiment in the absence of MTX than in its presence (10 nM). Please note that it would be impossible to detect a low Kd affinity using 500 nM of EGFP-VHH. In other words, can the authors show that the low microM Kd is not an apparent Kd that results from using a high concentration of EGFP-VHH? Most straightforward would be to repeat this titration with 10 nM EGFP-VHH as well.*

Response: The reviewer is correct that the experiment could have been done using a single low concentration of the reporter (EGFP-VHH), while varying concentrations of MTX and nanoCLAMP. However, the quality of the signal and the resulting data fit were worse than when we used higher concentration of the reporter. Given that the behaviour of the MTX biosensor based on their protein pair confirms a large affinity switch, we feel that we are unlikely to have made a significant error in estimating the affinity of the interaction.

Point 5 *I am a little confused by the statement that the authors can measure low nM concentrations of MTX in patient serum samples. In the SI it is mentioned that serum samples are diluted 40-4000-fold before they are assayed, so the authors cannot claim that they can do these assays in serum. Also, do de concentrations refer to the concentration in serum before dilution or the final concentration in the assay mixture? Experiments with patient serum samples were done using optical detection, which may*

explain the need for dilution. Did the authors test whether their final bioelectronic assay could be used with (diluted) serum samples?

The reviewer is correct that the serum samples were diluted. The concentrations shown in figure 4F refer to the concentration of the drug in the undiluted sample. We have now rewritten the section discussing the utility of the developed assays to clarify some of the points that the reviewer is raising. Please see the response to the point 2 of the reviewer 1.

Point 6 -*"The combination of high specificity with nanomolar sensitivity makes the proposed technology a potential contender to the leading methods of continues biochemical monitoring technologies."* -> *I think this not correct for two reasons. First, the assay still need reagents, which makes it challenging to be used for continuous monitoring.*

Response: We partially agree with the reviewer, as having a two component system (where one of the components is freely diffusing in solution) creates additional technical issues when applied to continuous monitoring. However, this can be overcome as described above in response 2 to referee 2.

Point 7 *Second, the performance of the sensor in complex medium such as undiluted serum has not be shown.*

Response: We have now included new data where we test performance of the developed MTX bioelectrodes in serum. These experiments demonstrate that the system can operate in at least 50% serum, and with further optimisation is likely to work in undiluted samples (Fig. 5F and Fig.S10E,F). This is not surprising given that the parental enzyme PQQ-glucose dehydrogenase is used in undiluted blood.

Reviewer #4

Point 1 *This work is an example of development of MTX sensor, while I suspect whether the demonstrated approach in this manuscript can be used for other small molecules than MTX. The author set title of the manuscript as generalized one as if the approach, CID, shown here can be applied for other small molecules.*

Response: This is a valid point, and we now expanded the discussion of methodologies for construction of CID systems (See response 3 to reviewer 1). We also further expanded the description of the experimental procedures and results to alert the readers to the fact that we recovered MTX-sensitive binders in two out of three targets. While development of CID systems is an area under active development, there has been a steady stream of reports (including ours) where selection systems have been successfully deployed to engineer well-functioning components.

Point 2 *The variable elements bind to beta-sheet of VHH and loop 64-71 binds to CDR1 and CDR4 of VHH and this binding moiety seems not as allosteric binding because conformational arrangements of CDRs 1 and 4 which are binding sites for the binder are induced by the recognition of MTX. Therefore molecular scheme depicted in figures including Fig4 seems to be misleading whereas I can agree with the authors' statement "the ligand induced conformational change creates an additional binding epitope for the nanoCLAMP" (P10 the bottom).*

Response: On advice of the referee we modified figure 4 and 5 to reflect the structural change in VHH domain that leads to an emergence of an additional binding interface.

Point 3 *Structural analysis of nanoCLAMP-GDH+MTX system was not shown which is necessary for supporting whether the outcome of the panning meets with the authors' idea including allosteric binding.*

Response: Given that the biosensor is composed of protein domains connected with flexible linkers, their crystallisation is most likely to be difficult. This complicates the use of X-ray crystallography and leaves cryo-EM as the only viable method. However, we feel that such a study is beyond the scope of the current paper, which is aimed at the development of CID systems and resulting biosensors. We have solved two independent structures of the VHH:MTX:nanoCLAMP complexes (the structures are now deposited to the PDB and accession numbers are provided), and believe that we provide sufficient amount of structural data to support our conclusions.

Point 4 *In fig.4C it looks that VHH was fused to GDH-CaM chimera and nanoCLAMP was fused to CaM-BP, while figure caption of fig4D explains that VHH-CaM-GDH and nanoCLAMP- CaMBP were used. This is quite confusing and difficult to understand completely.*

Response: This confusion is justified and we apologise for our mistake in annotating the figure (which has now been rectified).

Point 5 P4L10 why did the authors specify Fig1B here?

Response: Reference to Fig1B is redundant and was removed.

Point 6 P7L19 *"It is clear that the rearrangement of the CDR1 region (Ser199-W206) of VHH upon MTX binding plays a key role in selective binding of the nanoCLAMP." was not evident from Fig3C rather it seems that only slight change occurred.*

Response: There is a large conformational change from the apo- form (grey) to the MTX-bound VHH (blue). The reviewer probably was comparing the structure of VHH:MTX complex (blue) and the ternary VHH:MTX:nanoCLAMP complex (red).

Point 7 P7L15 *Although the authors stated "This arrangement is quite surprising as one would expect the selection to result in structures where the variable elements would interact with the ligand or its surrounding.", I do not have same impression since the authors did not explain the intended structural basis for the binder recognition of target (MTX-VHH complex in this case).*

Response: On page 4 (last paragraph) we described our experimental design where “we also focused on complexes where the small molecule was at least partially solvent exposed, thereby creating potential new binding modalities on the surface of the anchor domain”. Therefore, our initial intent was to design the screen in a way that would result in binders interfacing with the anchor domains via small molecular weight ligand. Hence our surprise is justified by the deviation of the experimental result from our initial expectation.

Point 8 *Calcium ion is found in the binder structure. however how did the authors control the involvement of calcium ion? Were there possibility of failure in getting positive clones because of lack in the calcium ion during the panning?*

Response: Nectagen routinely pans libraries of this scaffold in the absence of Calcium with a high success rate. It is speculated that the nanoCLAMP retains the bound calcium even when calcium is not supplied in the media, due to the presence of the calcium ion in the crystal structure solved in this work, as well as the original crystal structure, 2W1Q.

Point 9 Typos were found like P9L1 "revieald", need to be corrected.

Response: Typo is corrected.

Point 10 P13 Fig. 9B, C should be Figs. S9B, C

Response: The reference to the supplementary figure is corrected.

Reviewers' Comments:

Reviewer #1:

Remarks to the Author:

My main concerns with the original MS were that the novelty of the protein engineering approach was limited, and there were doubts as to the generalizability of the sensor design vis-à-vis detecting other analytes at point-of-care. With respect to the latter point, the authors have dialed down claims of generalizability and cite progress in development of CID systems by other groups.

The main interest in this paper boils down to the technical performance of the MTX sensor. I was a little confused about its intended use for therapeutic drug monitoring (TDM). For high-dose MTX cancer patients, the desired window for monitoring (toxic) concentrations appears to be 10-10,000 fold higher than K_d of the sensor (Howard et al., PMID 27496039), so serum from these patients would presumably have to be diluted by various amounts as in Fig. 4F. This is a minor niggle, but will this be a problem for continuous monitoring by the device that's envisioned? One might ideally want a version of the sensor with $K_d \sim 1 \mu\text{M}$ in this case, which should be relatively straightforward to design with the aid of the crystal structures. The authors state that their sensor can be useful for TDM in arthritis and Crohn's disease patients where therapeutic [MTX] is low nanomolar. But since this is orders of magnitude below toxic concentrations, is TDM actually useful for these patients?

Kobs (Fig. 4F and in the text) is a rate and should be denoted by lower case k. The fitted binding curve in Fig. 4A appears to be steeper than the curve in Fig. 4B. Were they fit to the same one-site equation?

Reviewer #2:

Remarks to the Author:

The revision is acceptably more careful and precise in its wording regarding, for example, the hurdles that remain before this approach can be made continuous. I wish the authors luck in surmounting those hurdles (if that is their interest).

Reviewer #3:

Remarks to the Author:

The authors have addressed almost all of my comments and questions satisfactorily. However, I still think that the authors have not convincingly demonstrated that they can measure low nM concentrations of MTX in patient serum samples. They provide two types of experiments on serum samples. The data shown in figure 4F were done using optical detection (not electrochemical detection) and thus required substantial dilution. In effect, the true LOD using this approach would be 100 nM MTX in serum samples. The authors have added new data on direct electrochemical detection of MTX in 50% serum (Figure 5F and S10E), but these data are very preliminary, showing only data for 150 and 300 nM, where the signal does not appear to depend linearly on the MTX concentration. In fact, the authors acknowledge that direct stable measurements in 50% serum are challenging (see also fig S10), which provides another argument to not oversell the performance of their current sensor (or alternatively provide the data to back up their claim of low nM sensitivity in serum).

Reviewer #4:

Remarks to the Author:

The authors properly responded to my questions and improved their manuscript sufficiently.

We addressed specific comments of the reviewers in the following way:

Reviewer #1

Point 1. *I was a little confused about its intended use for therapeutic drug monitoring (TDM). For high-dose MTX cancer patients, the desired window for monitoring (toxic) concentrations appears to be 10-10,000 fold higher than K_d of the sensor (Howard et al., PMID 27496039), so serum from these patients would presumably have to be diluted by various amounts as in Fig. 4F. This is a minor niggle, but will this a problem for continuous monitoring by the device that's envisioned? One might ideally want a version of the sensor with $K_d \sim 1 \mu\text{M}$ in this case, which should be relatively straightforward to design with the aid of the crystal structures.*

Response: These are important points, and we thank the reviewer for bringing them up. There are two possible modes in which biosensors can operate: the first is the active site titration where the absolute majority of the ligand added to the reaction forms a complex with the biosensor. This occurs under the condition where the concentration of the analyte is ca. >10 higher than the K_d of the analyte:biosensor complex. In this case the response remains linear until all of the biosensor is saturated. This mode of operation is predicated on the ability to construct biosensors with high affinity and very low background activity.

In an alternative approach, the biosensor is operated under equilibrium conditions and the concentration response range is determined by its K_d . One can shift or expand this range by changing the affinity of the binding domains for the analytes. As recognised by the referee, this is straightforward via the introduction of affinity-lowering mutations based on high resolution crystal structures.

We now focused on discussing the second approach and rearranged the section on page 17 to highlight the potential benefits provided by the developed biosensors to TDM of high-dose MTX treatment:

To demonstrate the practical utility of the developed artificial dimerization system we used them to develop GDH-based methotrexate biosensors, and show that the resulting solution-based assays could detect low nM concentrations of the drug in human serum with accuracy comparable to that of clinically used diagnostic methods. The sensitivity and simplicity of the assay allows its transfer from complex immunochemistry stations onto more simple and ubiquitous clinical chemistry analyzers and plate readers. Such tests can be performed in smaller diagnostic laboratories and community clinics, thus increasing access to MTX monitoring and improving treatment outcomes.

Our approach potentially allows construction of sensory arrays furnished with biosensor variants displaying different affinities for the analyte. Such an array could cover a large concentration range that otherwise would require serial dilution of the sample. In the case of MTX monitoring this would significantly simplify testing, as during cancer treatment the circulating MTX concentration can fluctuate over several orders of magnitude (10nM to >100µM) and display large (>5 fold) interindividual variability⁴⁰. Availability of high resolution structure of nanoCLAMP:MTX:VHH complex simplifies construction of biosensor variants with reduced affinities for MTX. Furthermore, a similar approach can be used to construct biosensor arrays capable of rapidly assessing condition-specific marker panels.

We also shifted the focus of the discussion on the potential use of the developed biosensor architecture for continuous monitoring away from specific applications in MTX monitoring and onto its potential broader utility.

One can also envisage that a similar approach can be utilized for construction of small molecule continuous monitoring systems that combine electrode immobilized GDH-CaM-Binder1 unit with soluble Binder2-CaM-BP. Such a system can be separated from the sample with a semi-permeable membrane, thus ensuring that the soluble component does not diffuse away and can associate with the electrode-bound reporter in a ligand-dependent manner. Use of such semi-permeable membranes in electrochemical biosensors is well established and would enable rapid transport of the analyte and circulating glucose to the biosensor⁴⁴. Although fluctuations of glucose levels will influence the accuracy of this approach, it can be overcome by either reducing the K_m of the GDH or by including a reference glucose electrode into the assembly.

Point 2. *The authors state that their sensor can be useful for TDM in arthritis and Crohn's disease patients where therapeutic [MTX] is low nanomolar. But since this is orders of magnitude below toxic concentrations, is TDM actually useful for these patients?*

Response: The reviewer makes a valid point as patients taking low doses of MTX for immune suppression are not typically monitored for MTX concentrations. While the treatment has side effects that may be in some cases severe, there is no consensus on the utility or best use of

TDM in those treatment protocols. There is clearly value in adherence monitoring where a simple PoC test would be of value. However, a discussion about the value of adherence monitoring is beyond the scope of this study. We therefore consolidated and modified our discussion on page 17 to focus on the high dose MTX treatment where TDM plays an integral part.

Point 3. *K_{obs}* (Fig. 4F and in the text) is a rate and should be denoted by lower case *k*.

Response: Corrected as advised

Point 4 *The fitted binding curve in Fig. 4A appears to be steeper than the curve in Fig. 4B. Were they fit to the same one-site equation?*

Response: We used the same equation for fitting but the experiments were done at different conditions of reactants in order to obtain a better quality of the signal.

Reviewer #3

Point 1 *I still think that the authors have not convincingly demonstrated that they can measure low nM concentrations of MTX in patient serum samples. They provide two types of experiments on serum samples. The data shown in figure 4F were done using optical detection (not electrochemical detection) and thus required substantial dilution. In effect, the true LOD using this approach would be 100 nM MTX in serum samples.*

Response: As can be seen in figure 4D, the detection limit of biosensor-based solution assay is < 0.5 nM of MTX. This means that dilution of serum sample containing MTX concentration 40-fold will allow drug quantification in the range of 10-20nM. To evidence that we now introduced figure S8 that shows the time resolved traces of the assay in the presence of serum samples containing MTX at concentrations 10 and 50nM. The sample containing 10 nM of MTX clearly separates from the background supporting our statement on page 17 (see above).

Point 2 *The authors have added new data on direct electrochemical detection of MTX in 50% serum (Figure 5F and S10E), but these data are very preliminary, showing only data for 150 and 300 nM, where the signal does not appear to depend linearly on the MTX concentration. In fact, the authors acknowledge that direct stable measurements in 50% serum are challenging (see also fig S10), which provides another argument to not oversell the*

performance of their current sensor (or alternatively provide the data to back up their claim of low nM sensitivity in serum).

Response: The reviewer is correct that we have not tested the developed MTX electrodes on samples containing single digit nanomolar concentrations of the drug. We believe that the current electrode design requires further improvement to detect such low concentrations in real biological samples. However, we do not claim that the presented electrode system achieves such sensitivity- such claims are made in respect to the colorimetric solution assay (see our response to point 1). The reviewer is correct that when electrode was exposed to 50% serum containing 0, 150 and 300nM concentration of MTX, the assay response to the last 2 concentrations was not linear. This is in accord with the data presented in figure 5E where these two concentrations map on to the last 20% of the response curve.

We believe we give the reader an accurate account of the electrode performance and its development stage as can be seen in the following statements:

P14: Further work will be required to optimize the electrochemical parameters of the assay to reduce the background signal, improve the efficiency of electron transfer, as well as find ways of tackling the high viscosity of the sample (which complicates work with undiluted serum samples).

P17: We demonstrate that such electrodes can generate significant and dose-dependent currents in response to analyte exposure. It is expected that the efficiency of the electron transfer, and hence the sensitivity of the system, can be further significantly improved by by attaching the biosensors to the electrode in an optimal orientation leading to improved electron transfer³⁸.

Reviewers' Comments:

Reviewer #1:

Remarks to the Author:

The responses to my concerns were adequate.

Reviewer #3:

Remarks to the Author:

I remain of the opinion that the authors oversell their claims regarding the performance of their system for electrochemical detection in serum. Instead of doing a proper titration experiment in Figure 5F with lower concentrations of MTX, they stick to showing preliminary data and rely on the argument that they acknowledge that the system should be improved in future experiments. However, one of the main statements in the abstract reads "We demonstrate utility of the developed CID by constructing electrochemical biosensors of methotrexate that enable accurate measurement of methotrexate in human serum." With this claim, it is up to the authors to provide a proper titration experiment over a wide concentration range with more than 3 data points. Apparently, the authors are not willing to do so, or alternatively adjust their claims.

We addressed the specific comments of the reviewer 3 in the following way:

Reviewer #3

Point 1 *I remain of the opinion that the authors oversell their claims regarding the performance of their system for electrochemical detection in serum. Instead of doing a proper titration experiment in Figure 5F with lower concentrations of MTX, they stick to showing preliminary data and rely on the argument that they acknowledge that the system should be improved in future experiments. However, one of the main statements in the abstract reads "We demonstrate utility of the developed CID by constructing electrochemical biosensors of methotrexate that enable accurate measurement of methotrexate in human serum." With this claim, it is up to the authors to provide a proper titration experiment over a wide concentration range with more than 3 data points. Apparently, the authors are not willing to do so, or alternatively adjust their claims.*

Response: We have now performed the requested MTX electrode titration experiment using full concentration range of the drug and the data is now presented in the figure 5F and Figure S10E. Comparing figures 5 E and F it can be seen that MTX electrode performs comparably in buffer and in the 50 % serum samples. We point the reader to the need for further optimisation to adopt them to undiluted samples. We also slightly changed the wording of the abstract to better reflect the state of technology development.

We demonstrate utility of the developed CID by constructing electrochemical biosensors of methotrexate that enable quantification of methotrexate in human serum.